# *Smchd1* is a maternal effect gene required for genomic imprinting

Iromi Wanigasuriya[1,2†], Quentin Gouil[1,2†], Sarah A Kinkel[1,2],
Andrés Tapia del Fierro[1,2], Tamara Beck[1], Ellise A Roper[3], Kelsey Breslin[1],
Jessica Stringer[4], Karla Hutt[4], Heather J Lee[3], Andrew Keniry[1,2],
Matthew E Ritchie[1,2,5], Marnie E Blewitt[1,2]*

[1]Walter and Eliza Hall Institute of Medical Research, Parkville, Australia; [2]The
Department of Medical Biology, The University of Melbourne, Parkville, Australia;
[3]Faculty of Health and Medicine, The University of Newcastle, Newcastle, Australia;
[4]Monash Biomedicine Discovery institute, Monash University, Clayton, Australia;
[5]The Department of Mathematics and Statistics, The University of Melbourne,
Parkville, Australia

**Abstract** Genomic imprinting establishes parental allele-biased expression of a suite of
mammalian genes based on parent-of-origin specific epigenetic marks. These marks are under the
control of maternal effect proteins supplied in the oocyte. Here we report epigenetic repressor
*Smchd1* as a novel maternal effect gene that regulates the imprinted expression of ten genes in
mice. We also found zygotic SMCHD1 had a dose-dependent effect on the imprinted expression of
seven genes. Together, zygotic and maternal SMCHD1 regulate three classic imprinted clusters and
eight other genes, including non-canonical imprinted genes. Interestingly, the loss of maternal
SMCHD1 does not alter germline DNA methylation imprints pre-implantation or later in gestation.
Instead, what appears to unite most imprinted genes sensitive to SMCHD1 is their reliance on
polycomb-mediated methylation as germline or secondary imprints, therefore we propose that
SMCHD1 acts downstream of polycomb imprints to mediate its function.

*For correspondence:
blewitt@wehi.edu.au

†These authors contributed
equally to this work

Competing interests: The
authors declare that no
competing interests exist.

Reviewing editor: Deborah
Bourc'his, Institut Curie, France

## Introduction

Genomic imprinting describes the process that enables monoallelic expression of a set of genes
according to their parent-of-origin (*McGrath and Solter, 1984*; *Surani et al., 1984*). Classically,
these imprinted genes are located in clusters. Disruption of imprinting at specific gene clusters leads
to imprinting disorders. For example, loss of imprinting at the *SNRPN* cluster is responsible for
Prader-Willi syndrome or Angelman syndrome, and loss of imprinting at the *IGF2/H19* and *KCNQ1*
clusters is responsible for Beckwith-Wiedemann syndrome (*Tucci et al., 2019*; *Peters, 2014*;
*Sutcliffe et al., 1997*; *Matsuura et al., 1997*; *Kishino et al., 1997*; *Nicholls et al., 1989*; *Sun et al.,
1997*; *Zhang et al., 1997*). Imprinted expression is enabled by germline DNA methylation or histone
methylation imprints that resist pre-implantation reprogramming and therefore retain their parent-
of-origin specific marks (*Ferguson-Smith, 2011*; *Barlow and Bartolomei, 2014*; *Inoue et al.,
2017a*; *Inoue et al., 2018*). The clusters controlled by germline DNA methylation imprints are classi-
cally described as the canonical imprinted genes. Whereas the more recently discovered imprinted
genes controlled by germline histone 3 lysine 27 trimethylation (H3K27me3) imprints have been
termed non-canonical imprinted genes (*Inoue et al., 2017a*; *Inoue et al., 2017b*). In addition to the
germline imprints, secondary imprints are found at some imprinted loci. For the canonical imprinted
genes, these can include both DNA methylation and histone methylation that occur post-fertilization
dependent on the germline imprint (*Hanna et al., 2019*).

Evidence suggests that proteins found in the oocyte establish both canonical and non-canonical imprints and enable them to resist pre-implantation reprogramming. Therefore, these maternally derived proteins are crucial for imprinted gene expression control. To date, only a small number of such maternal effect genes have been tested for and found to play a role in regulating imprinting, including *Trim28, Zfp57, Dppa3, Rlim, Nlrp2, Dnmt3l, Dnmt1,* and *Eed* (*Inoue et al., 2018*; *Inoue et al., 2017a*; *Howell et al., 2001*; *Payer et al., 2003*; *Hiura et al., 2006*; *Li et al., 2008*; *Shin et al., 2010*; *Messerschmidt et al., 2012*; *McGraw et al., 2013*; *Alexander et al., 2015*; *Mahadevan et al., 2017*).

SMCHD1 is an epigenetic modifier, the zygotic form of which is required for both X chromosome inactivation (*Blewitt et al., 2008*; *Gendrel et al., 2012*) and silencing of clustered autosomal loci, including genes at the *Snrpn* and *Igf2r-Airn* imprinted clusters (*Mould et al., 2013*; *Gendrel et al., 2013*; *Chen et al., 2015*; *Jansz et al., 2018a*). Homozygous removal of zygotic SMCHD1 causes the loss of imprinting at a few genes at these clusters, hypomethylation of the secondary DMR at the *Snrpn* cluster (*Mould et al., 2013*), but does not change the H3K27me3 found across the *Snrpn* imprinted region (*Chen et al., 2015*). SMCHD1 function is also relevant in the context of disease (*Jansz et al., 2017*). Heterozygous mutations in *SMCHD1* are found in two distinct human disorders: loss of function mutations in facioscapulohumeral muscular dystrophy (FSHD)(*Lemmers et al., 2012*), and potentially gain of function mutations in the rare craniofacial disorder Bosma arhinia and microphthalmia (BAMS)(*Gordon et al., 2017*; *Shaw et al., 2017*; *Gurzau et al., 2018*). The role of SMCHD1 in normal development and disease has led to an increase in interest in how and when it contributes to gene silencing at each of its genomic targets.

Recent work by our group and others have shown that SMCHD1 is required for long-range chromatin interactions on the inactive X chromosome (*Jansz et al., 2018a*; *Wang et al., 2018*; *Gdula et al., 2019*), and at its autosomal targets including the imprinted loci (*Jansz et al., 2018a*). In cells lacking SMCHD1, other epigenetic regulators are enriched at SMCHD1 targets, for example, CCCTC-binding factor (CTCF) binding is enhanced at the inactive X chromosome and the clustered protocadherins (*Chen et al., 2015*; *Wang et al., 2018*; *Gdula et al., 2019*). The inactive X also shows an accumulation of H3K27me3 in the absence of SMCHD1 (*Jansz et al., 2018a*). These data led to a model where SMCHD1-mediated chromatin interactions insulate the genome against the action of other epigenetic regulators, such as CTCF (*Chen et al., 2015*; *Jansz et al., 2018a*; *Wang et al., 2018*; *Gdula et al., 2019*); however, it is not known whether this model holds true for all SMCHD1 target genes.

Based on SMCHD1's known role at some imprinted clusters (*Mould et al., 2013*; *Gendrel et al., 2013*; *Chen et al., 2015*; *Jansz et al., 2018a*) and its expression in the oocyte (*Midic et al., 2018*), we sought to test the role of oocyte-supplied maternal Smchd1 in regulating imprinted gene expression. Here we show that *Smchd1* is a novel maternal effect gene, required for the imprinted expression of 10 genes. Furthermore, we reveal a dose-dependent role for zygotic SMCHD1 in regulating imprinted expression of seven genes across the *Snrpn*, *Igf2r-Airn*, and *Kcnq1* canonically imprinted clusters. What appears to unite most of these two sets of genes is the reliance on H3K27me3 germline or secondary imprints. These data lead us to propose that SMCHD1 interprets H3K27me3 imprints to establish an epigenetic state required for imprinted expression.

## Results

### Genetic deletion of maternal *Smchd1*

To determine whether maternally derived SMCHD1 is required for autosomal imprinting, we deleted *Smchd1* in the oocyte using MMTV-Cre or Zp3-Cre (*Figure 1a*). Contrary to previous reports where SMCHD1 depletion by siRNA knockdown in pre-implantation embryos resulted in reduced blastocyst formation, hatching and survival to term (*Midic et al., 2018*; *Ruebel et al., 2019*), we found no embryonic or preweaning lethality following the maternal deletion of *Smchd1* (*Figure 1—figure supplement 1a*), and normal weights at P2 (*Figure 1—figure supplement 1b*). Using MMTV-Cre, SMCHD1 was deleted progressively from the early secondary to antral follicle stages of oocyte development, whereas Zp3-Cre deleted earlier, in the primary follicle (*Figure 1b*). Using our SMCHD1-GFP fusion knock-in mouse (*Jansz et al., 2018a*), we found that paternally encoded native SMCHD1-GFP protein was not detectable until the 16-cell stage (*Figure 1c*). We confirmed the

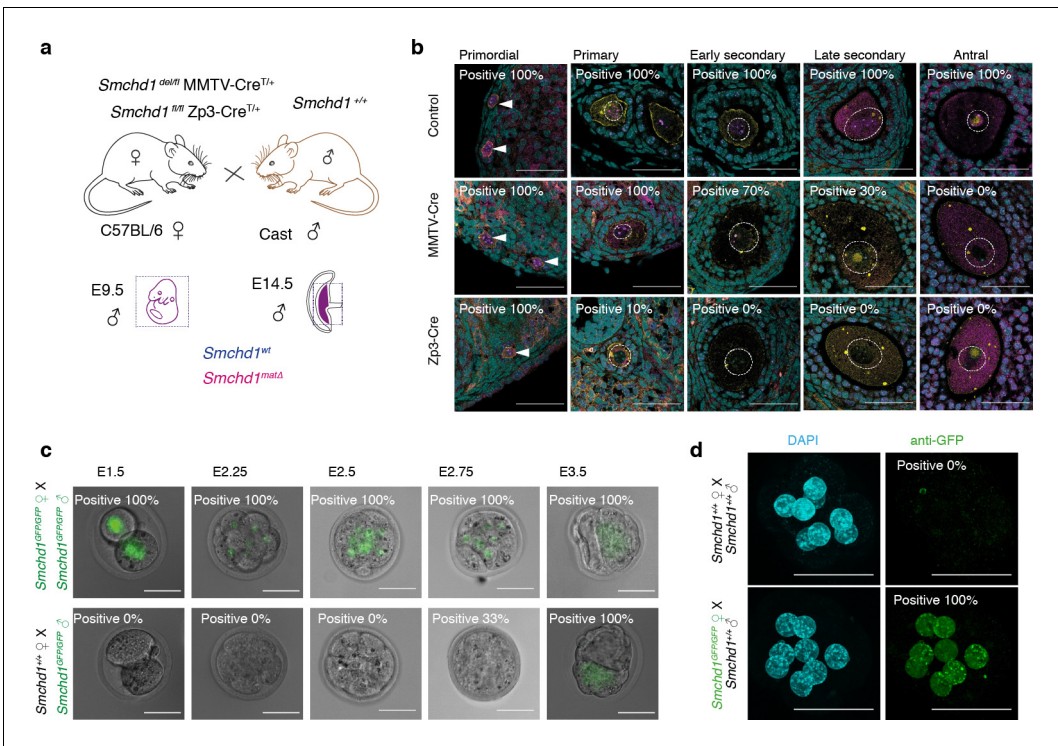

**Figure 1.** Maternal deletion of *Smchd1* during oocyte development depletes SMCHD1 until the 16-cell embryonic stage. (a) Schematic for maternal deletion of *Smchd1*. (b) Deletion of *Smchd1* in oocyte development with MMTV-Cre and Zp3-Cre. Arrowheads indicate primordial follicle oocyte nuclei, white dotted lines surround primary-antral follicle oocytes. Smchd1 (magenta), c-KIT (yellow), DAPI (cyan). n = 15–27 sections for two ovaries per cohort. A total of 5–20 follicles were observed for primordial – late secondary stages and 2–3 antral follicles for each genotype. (c) Detection of paternal SMCHD1-GFP from day 1.5 to 3.5 in pre-implantation embryos. *Smchd1*[GFP/GFP] embryos were used as positive controls. (d) Detection of maternal SMCHD1-GFP in the nuclei of E2.5 (8 cell) embryos, with *Smchd1*[+/+] used as negative controls. Nuclei marked with DAPI (cyan). Scale bar – 50 µm. The online version of this article includes the following figure supplement(s) for figure 1:

**Figure supplement 1.** Maternal deletion of *Smchd1* does not affect pup viability or weight.
**Figure supplement 2.** Minor global gene expression changes in *Smchd1*[matΔ] placentae and embryos.

expression by immunofluorescence and found very low levels of paternal SMCHD1 switching on from the 8-cell stage (*Figure 1—figure supplement 1c*), but its expression is much lower than maternal SMCHD1 at this time (*Figure 1d*). Together these data suggest maternal SMCHD1 is the primary source of protein until at least the 32-cell stage. These data define the period of maternal deletion of *Smchd1* in each model. Using the same approach, we also confirmed that maternal SMCHD1 localizes to the nucleus in the pre-implantation period (*Figure 1d*), as expected if SMCHD1 plays a functional role in gene silencing at this time.

Using this system, we analyzed the effect of maternal SMCHD1 on expression using RNA-seq in male *Smchd1* maternally deleted (*Smchd1*[matΔ]) E9.5 embryos and the embryonic portion of E14.5 placentae. We chose to sample embryos at E9.5 as this is before SMCHD1 zygotic-null females die; meanwhile mid-gestation placenta is the time and place where many genes display imprinted expression, including those known to be sensitive to loss of zygotic SMCHD1 (*Mould et al., 2013*; *Gendrel et al., 2013*; *Andergassen et al., 2019*). Females were not examined here due to the confounding role of SMCHD1 in X chromosome inactivation, which is the focus of a separate project. We found only five consistently differentially expressed genes between *Smchd1*[matΔ] and control placental samples from both Cre models, with modest fold-changes (*Figure 1—figure supplement 2a–c*, *Supplementary file 1–2*): one upregulated (*Gm8493*) and four downregulated (*Ceacam12*, *Psg16*, *Gm7863*, and *Afp*). Differential expression in the E9.5 embryos (MMTV-Cre, *Figure 1—figure supplement 2d*) was more pronounced (although not validated by a second Cre system), notably with

the four-fold upregulation of protocadherin genes (*Pcdhb17-22*) - known SMCHD1 targets (*Chen et al., 2015*). Overall there were 229 upregulated and 39 downregulated genes in this tissue (*Supplementary file 2*), most with modest fold-changes. They did not include imprinted genes and did not overlap with the misregulated genes common to the two Cre models in E14.5 placenta. These data suggest the removal of maternal SMCHD1 does not have a striking effect on global gene expression in the placenta and has a slightly larger effect in the embryo. These data are consistent with no observable effect of deleting maternal *Smchd1* on viability or postnatal weight (*Figure 1— figure supplement 1*).

## Maternal SMCHD1 is not required to regulate the imprinting of known SMCHD1 targets, although they are sensitive to *Smchd1* haploinsufficiency

Besides its documented role in silencing a number of autosomal clustered gene families such as protocadherins, zygotic SMCHD1 is also known to regulate a selection of imprinted genes. To examine the role of maternal SMCHD1 in imprinted expression we took advantage of the single nucleotide polymorphisms within the same C57BL/6 x CAST/EiJ F1 samples for allele-specific analyses (*Figure 1a*). We observed consistent effects on imprinted expression with both Cre strains, suggesting that the stage of deletion during oogenesis does not influence the outcome (*Figure 2—figure supplement 1a*). Therefore, we pooled data from both Cre strains for statistical analyses. We imposed a 5% FDR as well as an absolute difference in the proportion of the RNA derived from the silenced allele greater than 5% to call differential imprinted expression. E14.5 *Smchd1*^matΔ placentae displayed partial loss of imprinted expression at known SMCHD1-sensitive genes in the *Snrpn* cluster (*Magel2*, *Peg12*, and *Ndn*, *Figure 2a*, *Supplementary file 1*) and the *Igf2r-Airn* cluster (*Slc22a3 Figure 2b*, *Supplementary file 1*), due to an increase in the expression of the silenced allele (*Figure 2—figure supplement 1b and c*). We additionally observed that *Pde10a* displayed partial loss of imprinting in the SMCHD1 maternal null samples, adding this gene in the *Igf2r-Airn* cluster to the list of SMCHD1-sensitive gene (*Figure 2b*).

To test whether partial loss of imprinting was due to a maternal effect or haploinsufficiency for *Smchd1*, we created F1 *Smchd1* heterozygous E14.5 placentae that inherited the null allele from the sire, and littermate wildtype controls (*Figure 2—figure supplement 2a*). Similar to the maternal null samples, these showed virtually no global differential expression (only three genes passing below the 5% FDR, but with fold-changes below 2: downregulated *Smchd1* and upregulated *Peg12* and *Pisd-ps2*, *Figure 2—figure supplement 2b* and *Supplementary file 1*) as well as partial loss of imprinting at the same genes affected by the maternal deletion (*Figure 2a and b*).

In the embryo, far fewer genes are imprinted compared with the placenta, however partial loss of imprinted expression was also observed at *Peg12* and *Ndn* in E9.5 maternal null embryos (*Figure 2— figure supplement 2c*, *Supplementary file 1*).

Together these data show that the *Snrpn* and *Igf2r-Airn* clusters are sensitive to heterozygous loss of zygotic SMCHD1, consistent with previous findings that showed these genes exhibit loss of imprinting in zygotic null samples (*Mould et al., 2013*; *Gendrel et al., 2013*). At this stage of development, these clusters are not specifically sensitive to the loss of maternal SMCHD1.

We analyzed DNA methylation in male E14.5 placentae by reduced representation bisulfite sequencing (RRBS). Deletion of maternal *Smchd1* did not have a large effect on global CpG island methylation. Only 33 and 42 CpG islands out of 13,840 showed appreciable hypo- and hyper-methylation, respectively (*Figure 2—figure supplement 2d* and *Supplementary file 4*). The only association with differential expression was the hypermethylation of the CpG island 1.8 kb upstream of *Gm45104*, downregulated three-fold in the placental sample with MMTV-Cre *Smchd1* deletion.

Examining imprinted differentially methylated regions (DMRs), we found that despite partial loss of imprinted expression in *Smchd1*^matΔ samples, DNA methylation at germline DMRs at the *Snrpn* and *Igf2r-Airn* clusters was maintained (*Figure 2c*, *Supplementary file 3*). We and others previously showed that *Smchd1* null embryos display hypomethylation of the *Peg12* secondary DMR (*Mould et al., 2013*; *Gendrel et al., 2013*), however this region is not methylated in the placenta so we could not directly compare with our current data.

These data show that haploinsufficiency for *Smchd1* results in the partial loss of imprinted expression at the *Snrpn* and *Igf2r-Airn* clusters without altering imprinted DNA methylation. Maternal

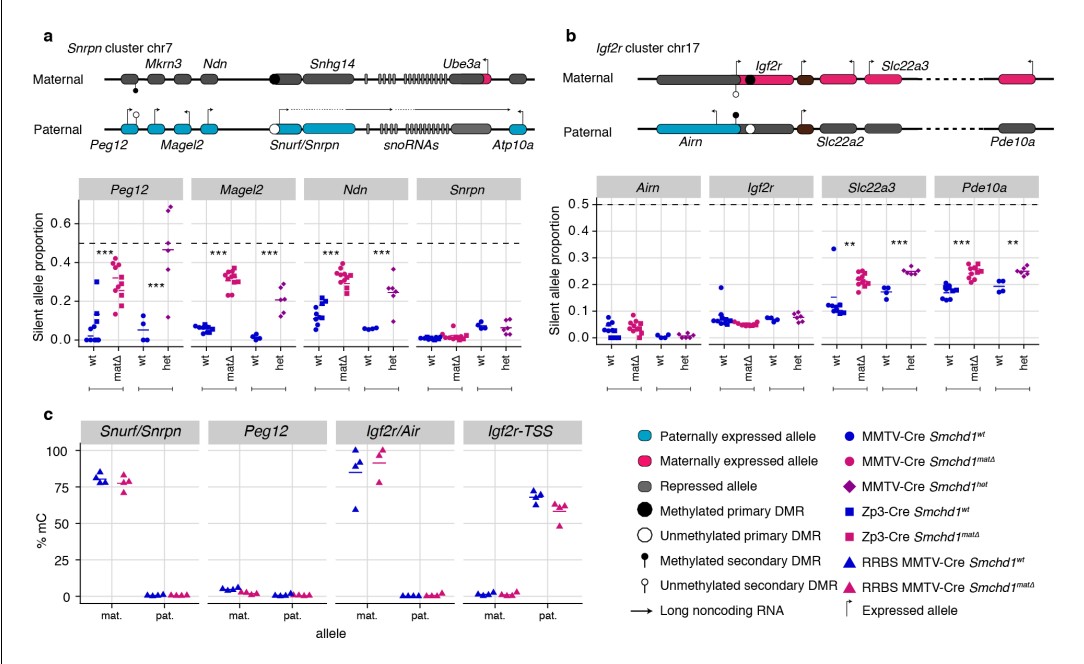

**Figure 2.** Heterozygous deletion of *Smchd1* results in partial loss of imprinting at known SMCHD1-sensitive clusters. (a-b) Expression of the silent allele as a proportion of total expression of the gene, obtained by allele-specific RNA-seq from the embryonic portion of the placenta of *Smchd1* wild-type (wt), heterozygous (het) and maternal null (matΔ) conceptuses. Expression data for (a) *Snrpn* cluster genes on chromosome seven and (b) *Igf2r* cluster genes on chromosome 17. (c) Percentage methylation (% mC) on the maternal and paternal alleles at primary and secondary DMRs at *Snrpn* and *Igf2r* clusters in *Smchd1* maternal null and wild-type placental samples. *p<0.05, **p<0.01, ***p<0.001, when the difference in mean silent allele proportions between genotypes is of at least 5%. RNA-seq sample sizes: for maternal deletion experiment, MMTV-Cre 6 wt and seven matΔ, Zp3-Cre 4 wt and four matΔ; for the heterozygous deletion experiment, 4 wt and six het E14.5 placental samples. RRBS: n = 4 MMTV-Cre for both matΔ and wt E14.5 placental samples.

The online version of this article includes the following figure supplement(s) for figure 2:

**Figure supplement 1.** Allele-specific gene expression changes in *Smchd1*^matΔ placentae and embryos.

**Figure supplement 2.** Global differential gene expression in *Smchd1* heterozygous (paternally deleted) placentae, *Snrpn* cluster allele-specific expression in *Smchd1*^matΔ embryos, and CpG island differential methylation in *Smchd1*^matΔ placentae.

SMCHD1 is not required for imprinted expression at this stage of development of genes in the *Snrpn* and *Igf2r-Airn* clusters where zygotic SMCHD1 plays a role.

## Two genes in the *Kcnq1* cluster are under the control of SMCHD1

Next, we analyzed all other canonical imprinted genes in both the maternal null and heterozygous placental samples. We observed partial loss of imprinted expression of two additional genes in the *Kcnq1* cluster, *Tssc4* and *Ascl2*. This change occurred without loss of imprinting at any other genes in the cluster, including the long non-coding RNA *Kcnq1ot1,* and the neighboring *Ckdn1c* and *Phlda2* genes whose silencing depends on allelic repressive histone marks. Similarly to the *Snrpn* and *Igf2r-Airn* clusters, we detected partial loss of imprinting in maternal null and heterozygous samples for *Ascl2*; however, *Tssc4* did not show a heterozygote effect, suggesting that maternal SMCHD1 may contribute to *Tssc4* imprinted expression (*Figure 3a*, *Supplementary file 1*). Again, consistent with the *Snrpn* and *Igf2r-Airn* clusters, we observed no DMR hypomethylation at either the germline (*Kcnq1ot1*) or secondary DMRs (*Kcnq1l1* and *2*) in the *Kcnq1* cluster (*Figure 3b*). Supporting a role for SMCHD1 in regulating the *Kcnq1* cluster, we found SMCHD1 enrichment across this cluster as well as the *Snrpn* and *Igf2r-Airn* imprinted clusters in placental samples and somatic cells (*Figure 3—figure supplement 1a–c*). These data reveal a broader role for SMCHD1 in canonical imprinting than previously known.

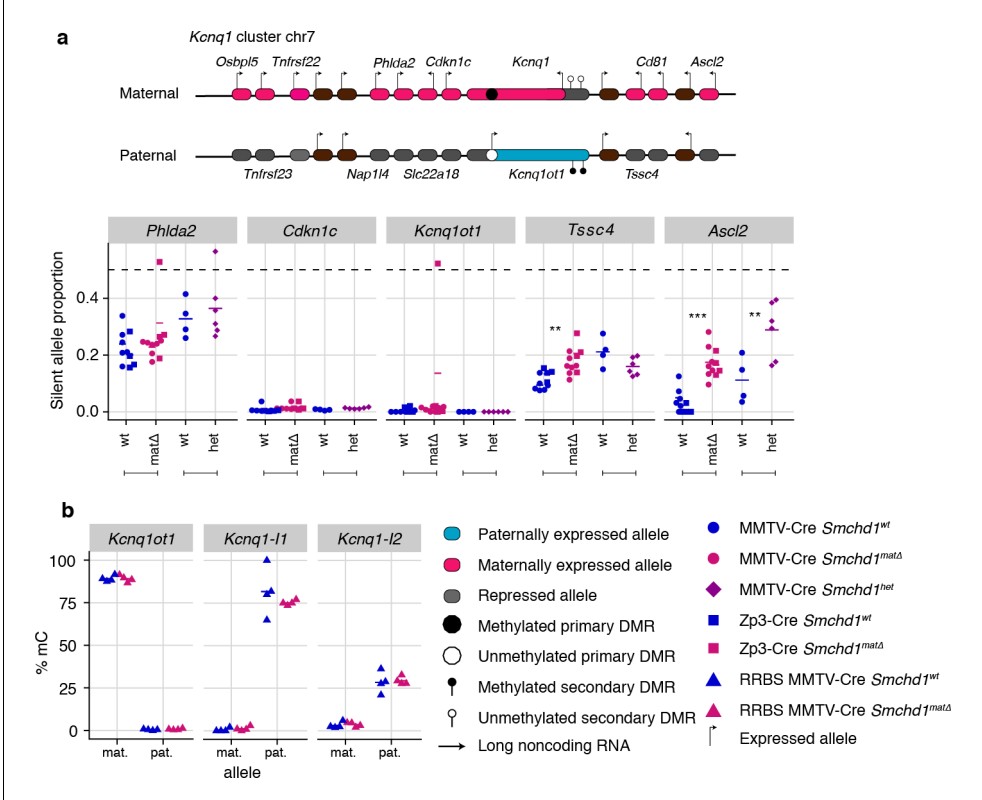

**Figure 3.** Maternal and heterozygous deletions of *Smchd1* result in loss of imprinting at the *Kcnq1* imprinted cluster without changes to primary or secondary DMR methylation. (a) Expression of the silent allele as a proportion of total expression of the gene, obtained by allele-specific RNA-seq from the embryonic portion of the placenta of *Smchd1* wild-type (wt), heterozygous (het), and maternal null (matΔ) conceptuses. Expression data for (a) *Kcnq1* cluster genes on chromosome 7, and (b) Percentage methylation (% mC) for each parental allele at the DMRs for the Kcnq1 cluster. *Kcnq1-I1: Kcnq1-Intergenic1; Kcnq1-I2: Kcnq1-Intergenic2* *p<0.05, **p<0.01, ***p<0.001, when the difference in silent allele proportions is of at least 5%. RNA-seq sample sizes: for maternal deletion experiment, MMTV-Cre 6 wt and seven matΔ, Zp3-Cre 4 wt and four matΔ; for the heterozygous deletion experiment, 4 wt and six het E14.5 placental samples. RRBS: n = 4 MMTV-Cre for both matΔ and wt E14.5 placental samples.

The online version of this article includes the following figure supplement(s) for figure 3:

**Figure supplement 1.** ChIP-seq for SMCHD1-GFP over the *Kcnq1* (a), *Snrpn* (b), and *Igf2r-Airn* (c) imprinted clusters in E14.5 placenta and neural stem cells.

## Maternal SMCHD1 regulates non-canonical imprinted gene expression

Using the same maternal null and heterozygous placental samples, we analyzed the effect of maternal SMCHD1 on the non-canonical imprinted genes, that have H3K27me3 germline imprints, and candidate new imprinted genes from our previous study (*Gigante et al., 2019*). In this case, we found that *Jade1, Platr4, Sfmbt2, Smoc1, Epop,* and *Spp1* showed partial loss of imprinted expression exclusively in the maternal null samples (*Tucci et al., 2019*; *Inoue et al., 2017b*; *Hanna et al., 2019*; *Matoba et al., 2018*; *Wang et al., 2011*; *Andergassen et al., 2017*; *Calabrese et al., 2015*). No loss of imprinting was observed for the other two non-canonical imprinted genes that retained imprinted expression at this stage of development (*Gab1* and *Slc38a4*) (*Inoue et al., 2017b*). No loss of imprinting was seen in heterozygous samples (*Figure 4a*, *Supplementary file 1*), suggesting that maternal SMCHD1 is required for appropriate imprinted expression of these non-canonical imprinted genes.

While the non-canonical imprinted genes have germline H3K27me3, they sometimes also have a secondary DNA methylation imprint, such as at *Sfmbt2* and *Jade1*. We observed no DMR

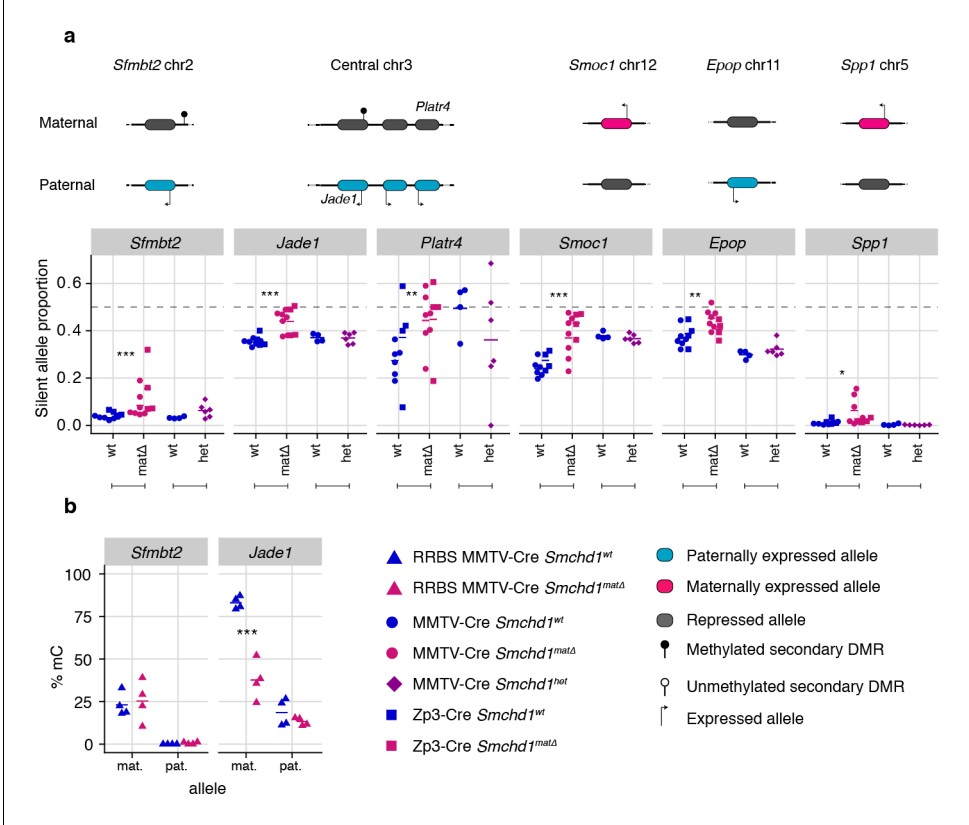

**Figure 4.** Only maternal deletion of *Smchd1* results in loss of imprinting at lone imprinted genes. (a) Expression of the silent allele as a proportion of total expression of the gene, obtained by allele-specific RNA-seq from the embryonic portion of the placenta of *Smchd1* wild-type (wt), heterozygous (het), and maternal null (matΔ) conceptuses, at lone imprinted genes: *Sfmbt2, Jade1, Platr4, Smoc1, Epop* and *Spp1*. (b) Percentage methylation (% mC) for each parental allele at the DMRs for *Sfmbt2* and *Jade1*. *p<0.05, **p<0.01, ***p<0.001, when the difference in silent allele proportions is of at least 5%. RNA-seq sample sizes: for maternal deletion experiment, MMTV-Cre 6 wt and seven matΔ, Zp3-Cre 4 wt and four matΔ; for the heterozygous deletion experiment, 4 wt and six het E14.5 placental samples. RRBS: n = 4 MMTV-Cre for both matΔ and wt E14.5 placental samples.

hypomethylation at *Sfmbt2*, but we did observe hypomethylation at the *Jade1* secondary DMR (*Figure 4b*, *Supplementary file 3*). These data suggest that maternal SMCHD1 can exert its effect without affecting the secondary DMR methylation.

## Maternal and zygotic SMCHD1 contribute to non-canonical imprinted gene expression

Earlier work on *Zfp57* and *Trim28* showed that either maternal or zygotic deletions resulted in partially penetrant loss of imprinting, whereas deletion of both maternal and zygotic increased the penetrance, evidence that both the oocyte and zygotic supply of these proteins contribute to imprint regulation (*Li et al., 2008*; *Alexander et al., 2015*). To understand the contribution of maternal versus zygotic SMCHD1 at imprinted genes, we produced embryonic and placental samples that were wild-type, *Smchd1*^matΔ, zygotic-deleted (*Smchd1*^zygΔ) or maternal-and-zygotic-deleted for *Smchd1* (*Smchd1*^matzygΔ). The consequence of our specific breeding scheme was that on average only half of the imprinted genes in each sample had the polymorphisms between parental alleles required for allele-specific analyses (*Figure 5—figure supplement 1a*), and some genes had no informative samples. Acknowledging this limitation, we only identified one additional SMCHD1-sensitive imprinted genes in *Smchd1*^zygΔ or *Smchd1*^matzygΔ samples compared with the maternal deletion alone: *Mkrn3* in the *Snprn* cluster in the embryonic samples. Although not significant, the maternal deleted

samples were consistent with a partially penetrant loss of imprinting at this gene as well (*Supplementary file 1*, *Figure 5—figure supplement 1b*).

In general, we observed a stronger effect of homozygous zygotic deletion on imprinted gene expression compared with maternal *Smchd1* deletion or heterozygosity for *Smchd1* (*Figure 5a–c*, *Figure 5—figure supplement 1b and c*). For the canonical imprinted genes, *Peg12*, *Magel2*, *Mkrn3*, and *Ndn* showed complete loss of imprinted expression in *Smchd1^zygΔ^* and *Smchd1^matzygΔ^* embryos and placentae, compared with only partial loss of imprinted expression in *Smchd1^matΔ^* or heterozygous samples (*Figure 2a*, *Figure 5a*, and *Figure 5—figure supplement 1b*). *Slc22a3* and *Pde10a* *Smchd1^zygΔ^* samples also showed significantly higher loss of imprinted expression compared to *Smchd1^matΔ^* or heterozygous samples in placentae (*Figure 5b*). At the *Kcnq1* cluster, for the samples

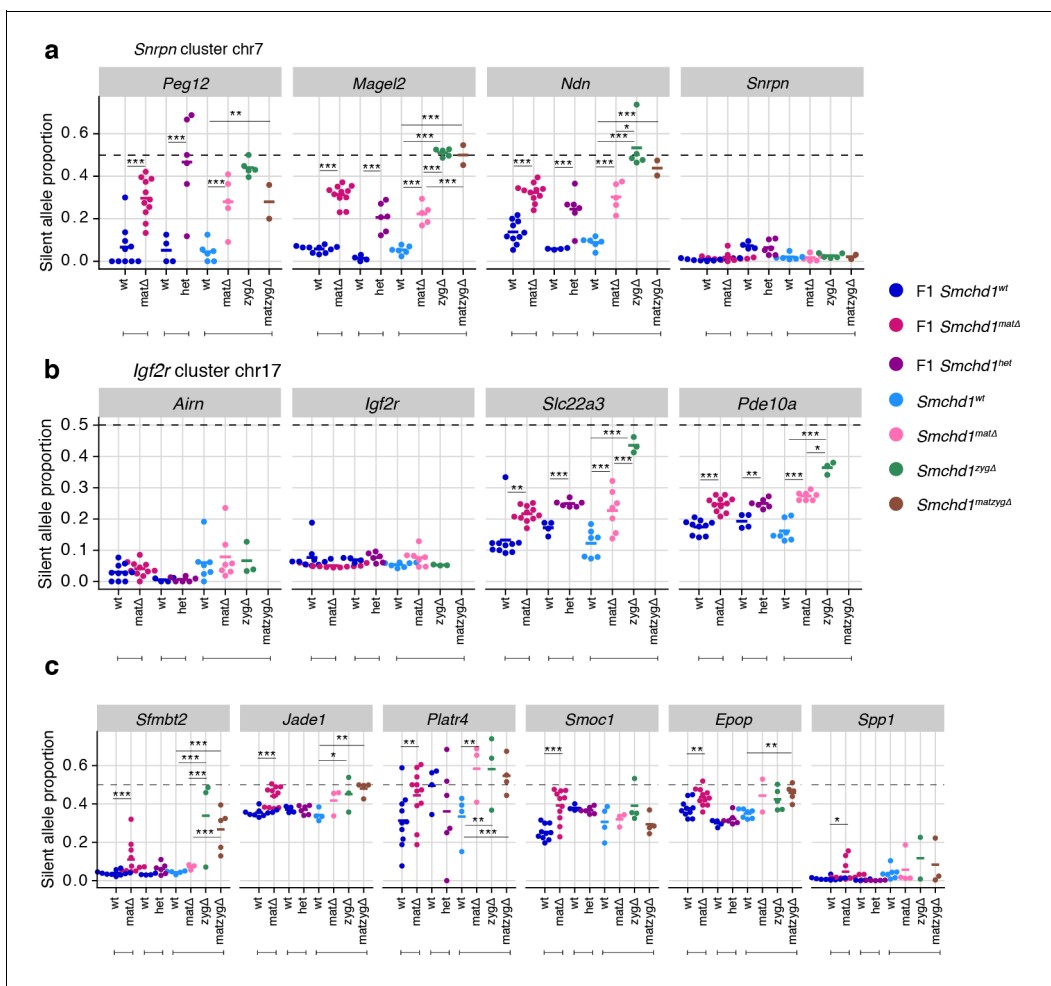

**Figure 5.** Homozygous zygotic deletion of *Smchd1* generally results in more severe loss of imprinting at genes sensitive to maternal or heterozygous deletions. MMTV-Cre *Smchd1* maternal deletion data and heterozygous deletion data (F1 wt, F1 matΔ, F1 het) from *Figures 1* and *2*, along with samples produced to compare *Smchd1* wild-type (wt), oocyte-deleted (matΔ), zygote-deleted (zygΔ), and oocyte-and-zygote-deleted (matzygΔ) genotypes. Samples from the embryonic portion of the placenta and expression of the silent allele is shown as a proportion of total expression of the gene, obtained by allele-specific RNA-seq. (a) *Snrpn* cluster genes. (b) *Igf2r-Airn* cluster genes. (c) *Sfmbt2*, *Jade1*, *Platr4*, *Smoc1*, *Epop*, and *Spp1* genes. *p<0.05, **p<0.01, ***p<0.001, when the difference in silent allele proportions is at least 5%. RNA-seq sample sizes: for maternal deletion experiment, 10 wt and 11 matΔ; for the heterozygous deletion experiment, 4 wt and six het; for the maternal and zygotic deletion experiment, 13 wt, seven matΔ, eight zygΔ, and six matzygΔ E14.5 MMTV-Cre placentae.

The online version of this article includes the following figure supplement(s) for figure 5:

**Figure supplement 1.** Allele-specific expression in maternal, zygotic, and maternal-and-zygotic *Smchd1*-deleted samples.

we had, we observed stronger loss of imprinting at *Tssc4* and *Ascl2* in samples with a zygotic and maternal deletion of *Smchd1*. Since we do not have zygotic deletion alone, we cannot confirm if this is due to solely zygotic SMCHD1 or a combination of maternal and zygotic SMCHD1. Together, these data are consistent with zygotic SMCHD1 playing the primary role at these canonical imprinted genes.

In general, we did not observe enhanced loss of imprinting at non-canonical imprinted genes from the zygotic null placentae. Notably, the non-canonical imprinted genes were not all strongly imprinted in the placental samples, as expected based on earlier publications (*Inoue et al., 2017b*; *Hanna et al., 2019*), and this potentially limited our ability to detect stronger loss of imprinting. *Sfmbt2* was the exception, where we both observed strong imprinted expression and significantly more loss of imprinting in the *Smchd1*$^{zyg\Delta}$ and *Smchd1*$^{matzyg\ \Delta}$ compared with *Smchd1*$^{mat\ \Delta}$ placental samples (*Figure 5c*). These data suggest that both maternal Smchd1 and zygotic SMCHD1 can play a role in regulating some non-canonical imprinted genes.

## Maternal SMCHD1 sets up imprinting during the pre-implantation period

Given that maternal SMCHD1 plays its role in the pre-implantation period, we directly tested the role of maternal SMCHD1 using allele-specific methylome and transcriptome sequencing of *Smchd1*$^{mat\Delta}$ and control 16 cell stage embryos. This is a period when maternally derived SMCHD1 provides the primary supply of SMCHD1 protein (*Figure 1c*). Although we found minimal differential expression genome-wide (*Figure 6—figure supplement 1a*), we found 89 genes with imprinted expression at this stage (*Supplementary file 1*), four of which showed significant loss of imprinted expression in the *Smchd1*$^{mat\Delta}$ samples (*Figure 6a and b*, *Figure 6—figure supplement 1b and c*). Two out of the four genes, *Jade1* and *Etv6*, were non-canonical imprinted genes; *Jade1* also showed loss of imprinting in the placental samples but *Sfmbt2* did not yet show an effect. These data suggest SMCHD1 is already necessary for imprinting of *Jade1* and at least one more non-canonical imprinted gene in the pre-implantation period, but by contrast is required for maintenance of imprinted expression for *Sfmbt2* (*Figure 6a and b*).

Unfortunately, none of the other SMCHD1-sensitive imprinted genes identified later in gestation were able to be analyzed for differential imprinted expression at this time based on both expression and coverage of informative SNPs. Therefore, we were unable to assess the role of maternal SMCHD1, particularly at canonical imprinted genes where zygotic SMCHD1 plays a role, at this stage of development.

An alternative way to look at canonically imprinted genes is to assess the effect of maternal SMCHD1 on germline DNA methylation imprints. Our previous DNA methylation analyses were performed on samples taken 11–12 days after zygotic SMCHD1 was switched on, potentially masking any effect on DNA methylation that could have occurred earlier in development. Therefore, we chose to analyze DNA methylation at E2.75 (16 cell stage), before zygotic SMCHD1 has made a contribution. We again found no differences at germline DMRs. As expected, secondary DMRs were not appreciably methylated at this stage (*Figure 6c*, *Supplementary file 3*). No change in germline DMR methylation is consistent with the (*Chen et al., 2015*; *Jansz et al., 2018a*) loss of imprinting only occurring at selected genes within the *Snrpn*, *Kcnq1*, and *Igf2r-Airn* clusters. These data show that maternal SMCHD1 does not contribute to DNA methylation at germline DMRs. Furthermore, we observe little genome-wide differential methylation in the pre-implantation samples: only 8 out 13,840 CpG islands, 45 out of 52,444 promoters, and 872 out of 272,566 10 kb bins were differentially methylated, suggesting that maternal SMCHD1 contributes little to overall methylation patterns at the pre-implantation stage (*Figure 6—figure supplement 1d-f* and *Supplementary file 4*), similar to later in gestation.

## Discussion

Here, we identified *Smchd1* as a novel maternal effect gene. For the first time, we revealed new canonical and non-canonical imprinted genes that are sensitive to deletion of maternal SMCHD1 (*Table 1*). These genes are also regulated by zygotic SMCHD1. We also revealed a third cluster of canonical imprinted genes regulated by zygotic SMCHD1, the *Kcnq1* cluster. The three canonical imprinted clusters have a dose-dependent reliance on zygotic SMCHD1, unlike the non-canonical

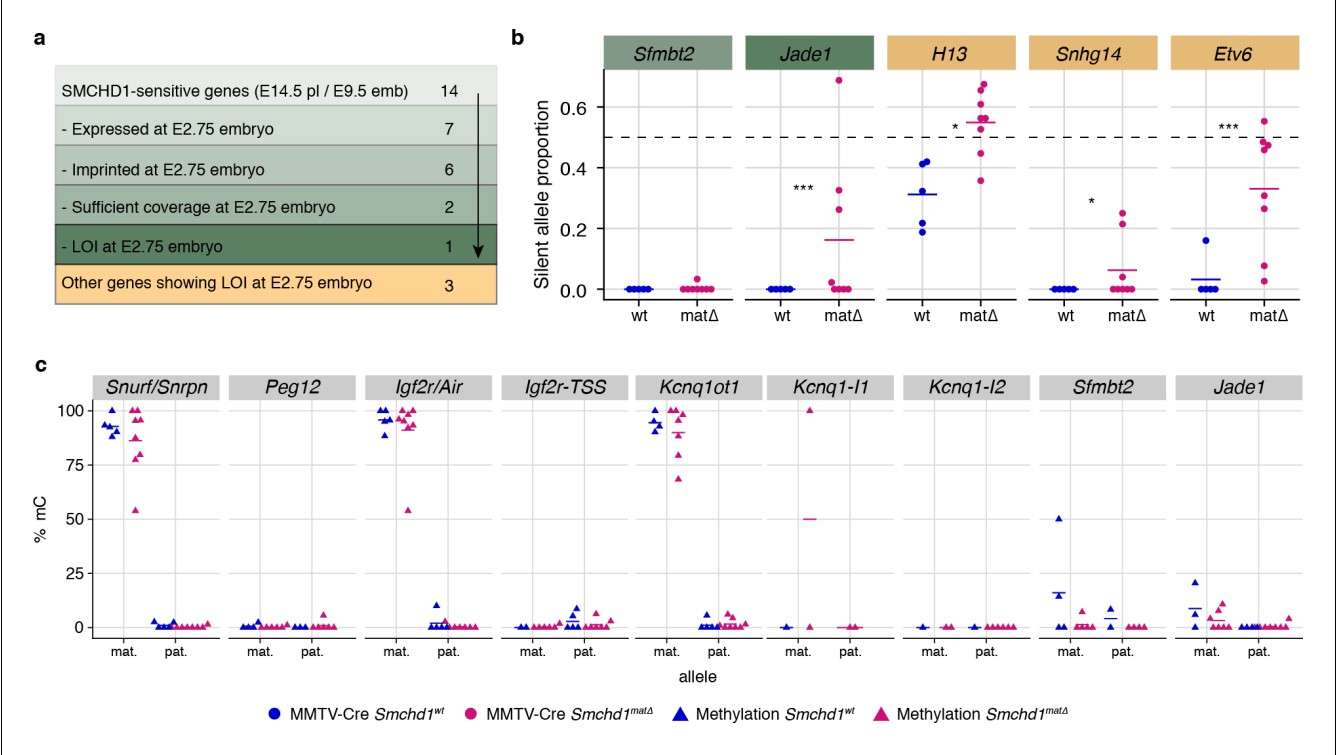

**Figure 6.** Maternal SMCHD1 establishes an epigenetic memory required for imprinted gene expression. (a) Summarised analysis of SMCHD1-sensitive imprinted genes from E14.5 placentae and E9.5 embryos in E2.75 embryo transcriptome sequencing. (b) Expression of the silent allele as a proportion of total expression of the gene, obtained by allele-specific RNA-seq from whole E2.75 *Smchd1* maternal null (matΔ) and wild-type (wt) embryos. (c) Percentage methylation (% mC) on each parental allele for the DMRs of SMCHD1-sensitive imprinted clusters and genes in *Smchd1* maternal null (matΔ) compared with control (wt) E2.75 embryos. *Kcnq1-I1: Kcnq1-Intergenic1; Kcnq1-I2: Kcnq1-Intergenic2* *p<0.05, **p<0.01, ***p<0.001, when the difference in silent allele proportions is at least 5%. n = 5 wt and n = 8 matΔ E2.75 embryos.

The online version of this article includes the following figure supplement(s) for figure 6:

**Figure supplement 1.** MA-plots of total (a) and allelic (b,c) gene expression in MMTV-Cre maternal deletion E2.75 embryos experiments.

imprinted genes. Together these data dramatically expand the number of imprinted loci that are sensitive to SMCHD1 removal.

For some non-canonical imprinted genes (*Smoc1*, *Jade1*, *Sfmbt2*), it has recently been shown that the H3K27me3 imprint is lost during pre-implantation development and through an unknown mechanism the imprint is transferred into a secondary DNA methylation imprint (*Hanna et al., 2019*). By contrast, the *Kcnq1* cluster, the *Igf2r-Airn* cluster and the *Magel2* region of the *Snrpn* cluster gain H3K27me3 in post-implantation tissues, dependent on the germline DMR imprint (*Hanna et al., 2019*), and for the *Kcnq1* and *Igf2r-Airn* clusters through the action of imprinted long noncoding RNAs (*Redrup et al., 2009*; *Nagano et al., 2008*; *Sleutels et al., 2002*). Interestingly, *Dnmt1* knockout studies have shown that while imprinting of genes located centrally in the *Kcnq1* cluster is maintained via germline DMR methylation, genes affected by the loss of SMCHD1, *Tssc4*, and *Ascl2*, are maintained via histone modifications (*Frost and Moore, 2010*; *Lewis et al., 2004*; *Umlauf et al., 2004*). Similar to the distal *Tssc4* and *Ascl2* genes in the *Kcnq1* cluster, the SMCHD1-sensitive genes at the *Snrpn* and *Igf2r-Airn* clusters are also far away from the ICR, while the genes whose promoter or TSS contain the ICR are not affected by the loss of SMCHD1. What appears to unite the SMCHD1-sensitive imprinted genes is their reliance on H3K27me3 either as a germline or secondary imprint. Based on two lines of evidence, we suggest that SMCHD1 acts downstream of these H3K27me3 imprints. First, we find the most striking loss of imprinted expression at both non-canonical and canonical imprinted genes in *Smchd1*ᶻʸᵍᐞ samples (*Figure 4*). These samples likely have no

**Table 1.** Summary of SMCHD1-sensitive imprinted genes.

| gene | cluster | type | sensitive tissue | Smchd1 sensitivity | haploinsufficiency |
|------|---------|------|------------------|--------------------|--------------------|
| Peg12 | Snrpn | canonical | E9.5 embryo, E14.5 placenta | zygotic | yes |
| Magel2 | Snrpn | canonical | E9.5 embryo, E14.5 placenta | zygotic | yes |
| Ndn | Snrpn | canonical | E9.5 embryo, E14.5 placenta | zygotic | yes |
| Mkrn3 | Snrpn | canonical | E9.5 embryo, E14.5 placenta | zygotic | not significant |
| Snhg14 | Snrpn | canonical | E2.75 embryo | maternal | no |
| Slc22a3 | Airn/Igf2r | canonical | E14.5 placenta | zygotic | yes |
| Pde10a | Airn/Igf2r | canonical | E14.5 placenta | zygotic | yes |
| Tssc4 | Kcnq1 | canonical | E14.5 placenta | maternal and zygotic | no |
| Ascl2 | Kcnq1 | canonical | E14.5 placenta | zygotic | yes |
| Jade1 | Jade1 | non-canonical | E2.75 embryo, E14.5 placenta | maternal | no |
| Platr4 | Jade1 | non-canonical | E14.5 placenta | maternal | no |
| Sfmbt2 | lone | non-canonical | E14.5 placenta | maternal | no |
| Smoc1 | lone | non-canonical | E14.5 placenta | maternal | no |
| Epop | lone | ? | E14.5 placenta | maternal | no |
| Spp1 | lone | ? | E14.5 placenta | maternal | no |
| H13 | lone | canonical | E2.75 embryo | maternal | no |
| Etv6 | lone | non-canonical | E2.75 embryo | maternal | no |

disruption to SMCHD1 levels in the oocyte or pre-implantation and therefore the non-canonical H3K27me3 imprints should remain. Second, we have previously found no effect on H3K27me3 in Smchd1 zygotic null samples at the SMCHD1-sensitive clustered imprinted genes, that is, secondary H3K27me3 imprints (*Chen et al., 2015*). Together these data suggest SMCHD1 acts downstream of H3K27me3 in mediating imprinted expression (*Figure 6*). We have previously shown that for the inactive X chromosome, SMCHD1 is recruited in a polycomb repressive complex 1-dependent manner (*Jansz et al., 2018b*). Potentially, a related pathway exists at these imprinted clusters.

We propose that SMCHD1 plays a role in protecting the repressed imprinted genes from inappropriate activation. We and others have previously found that cells without SMCHD1 have increased CTCF binding (*Chen et al., 2015*; *Wang et al., 2018*; *Gdula et al., 2019*). At SMCHD1-sensitive imprinted genes we have found increased H3K4me2 or H3K4me3 (*Mould et al., 2013*; *Chen et al., 2015*). Therefore, we suggest that SMCHD1 may ensure that H3K27me3 imprints result in transcriptional silencing by preventing the action of CTCF or other epigenetic activators via an insulating mechanism.

## Conclusions

We have discovered SMCHD1 is a novel maternal effect gene required for genomic imprinting. Recently, we and others discovered that zygotic SMCHD1 mediates long-range chromatin interactions at its target genes, both at imprinted regions and elsewhere in the genome (*Jansz et al., 2018a*; *Wang et al., 2018*; *Gdula et al., 2019*). Based on the data presented here, we propose that SMCHD1 is required to translate germline imprints into a chromatin state required to silence expression at select imprinted genes. Given that some SMCHD1-sensitive imprinted genes are not yet expressed in the pre-implantation period, we hypothesize the chromatin state that maternal SMCHD1 establishes, provides an epigenetic memory that is then maintained by zygotic SMCHD1 (*Figure 7*). This work opens a new avenue to understand how imprinted expression is established during development and may be relevant for patients with *SMCHD1* mutations.

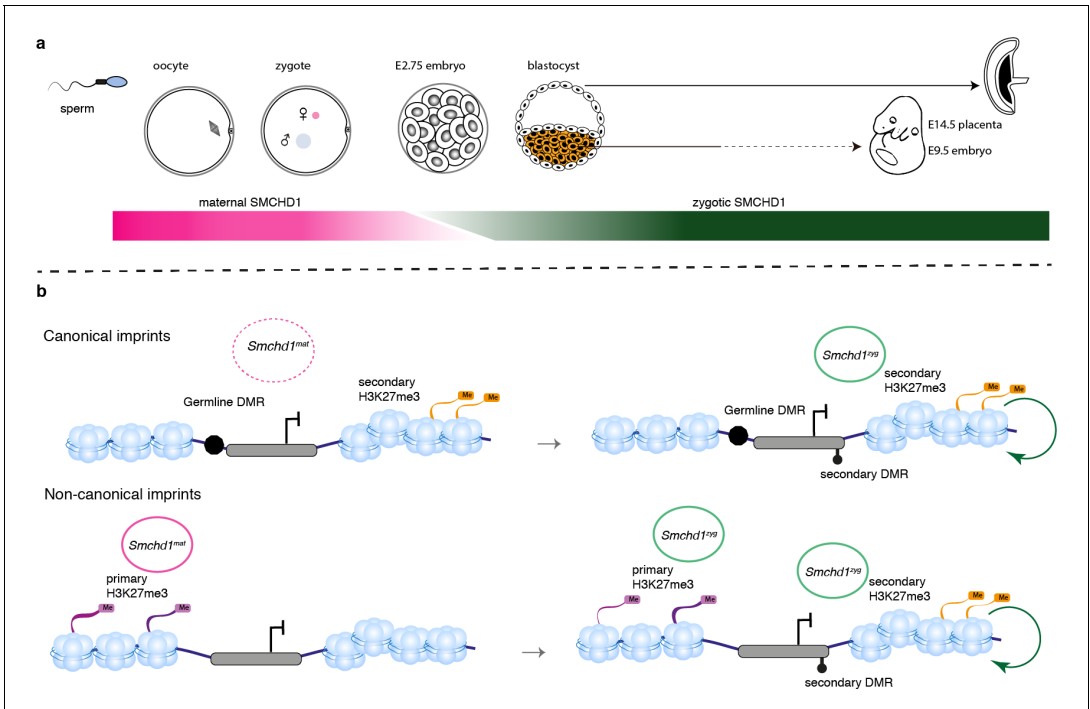

**Figure 7.** SMCHD1 translates the imprints to establish a heritable chromatin state required for imprinted expression later in development. (**a**) Developmental windows of activity of maternal and zygotic SMCHD1. (**b**) Proposed model illustrating the regulation of imprinted genes by SMCHD1. Both oocyte and zygotic SMCHD1 contribute to an epigenetic memory downstream of polycomb repressive histone marks.

# Materials and methods

## Key resources table

| Reagent type (species) or resource | Designation | Source or reference | Identifiers | Additional information |
|---|---|---|---|---|
| Gene (*M. musculus*) | Structural maintenance of chromosomes hinge domain containing 1 (*Smchd1*) | PMID:15890782 PMID:18425126 | NM_028887 | |
| Strain, strain background (*M. musculus*, males and females) | Castaneus EiJ | The Jackson Laboratory | 000928 | |
| Strain, strain background (*M. musculus*, females and males) | C57BL/6J | The Jackson Laboratory | 000664 | |
| Genetic reagent (*M. musculus*, females and males) | MMTV-Cre | PMID:9336464 | | Strain back-crossed from FVB/N to C57BL/6J background for this study |
| Genetic reagent (*M. musculus*, females and males) | Zp3-Cre | PMID:9016703 | | C57BL/6J background |

*Continued on next page*

*Continued*

| Reagent type (species) or resource | Designation | Source or reference | Identifiers | Additional information |
|---|---|---|---|---|
| Genetic reagent (*M. musculus*, females and males) | Smchd1 conditional knockout (*Smchd1$^{fl}$*) | PMID:29281018 | | C57BL/6J background |
| Genetic reagent (*M. musculus*, females and males) | *Smchd1$^{fl/fl}$* MMTV-Cre | This study | | C57BL/6J background. MMTV-Cre transgene on the *Smchd1$^{fl}$* background. |
| Genetic reagent (*M. musculus*, females and males) | *Smchd1$^{fl/fl}$* Zp3-Cre | This report | | C57BL/6J background, Zp3-Cre transgene on the *Smchd1$^{fl/fl}$* background. |
| Genetic reagent (*M. musculus*, males and females) | *Smchd1$^{GFP}$* | PMID:30127357 | | C57BL/6J background |
| Biological sample (*M. musculus*, male) | *Smchd1$^{GFP/GFP}$*, primary Neural Stem Cells (NSCs) | This study | | Produced fresh from *Smchd1$^{GFP/GFP}$* E14.5 embryos in Blewitt lab. |
| Antibody | anti-SMCHD1 ATPase domain (rat monoclonal) | In house. Source - PMID:30127357 | #5 or Clone 5H4 | (1:100) |
| Antibody | anti-SMCHD1 ATPase domain (rat monoclonal) | In house. Source - PMID:30428357 | #8 or Clone 2B8 | (1:100) |
| Antibody | anti-c-Kit (goat polyclonal) | Novus | Cat. #: af1356 | (1:500) |
| Antibody | anti-GFP (rabbit polyclonal) | Thermo Fisher Scientific | Cat. #: A11122 | (1:100) |
| Antibody | anti-goat Alexa 488 (donkey polyclonal) | Thermo Fisher Scientific | Cat. #: A-11055 | (1:500) |
| Antibody | anti-rat IgG DyLight 550 (donkey polyclonal) | Invitrogen | Cat. #:SA5-10027 | (1:500) |
| Antibody | anti-rabbit Alexa 488 (donkey polyclonal) | Thermo Fisher Scientific | Cat. #: A21206 | (1:500) |
| Chemical compound, drug | ProLong Diamond Antifade Mountant with DAPI | Thermo Fisher Scientific | Cat. #: P36931 | |
| Chemical compound, drug | Vectashield | Vector Labs | H-1000 | |
| Commercial assay or kit | Qiagen All prep kit | Qiagen | Cat. #:80204 | |
| Commercial assay or kit | Zymo Quick DNA/RNA miniprep plus kit | Zymo research | Cat. #:D7003 | |
| Commercial assay or kit | TruSeq V1 or V2 RNA sample preparation kit | Illumina | Cat. #:RS-122–2001 Cat. #:RS-122–2002 | |
| Commercial assay or kit | Zymo research DNA Clean and concentrator-5 kit | Zymo research | Cat. #:D4103 | |

*Continued on next page*

*Continued*

| Reagent type (species) or resource | Designation | Source or reference | Identifiers | Additional information |
|---|---|---|---|---|
| Commercial assay or kit | Qubit dsDNA assay kit | ThermoFisher Scientific | Cat. #:Q32853 | |
| Commercial assay or kit | NuGEN Ovation RRBS methyl-seq system | Integrated sciences | Cat. #:0553–32 | |
| Commercial assay or kit | QIAGEN EpiTect Fast DNA Bisulfite Kit | Qiagen | Cat. #:59824 | |
| Commercial assay or kit | Illumina TruSeq DNA Sample Preparation Kit | Illumina | Cat. #:FC-121–2001 Cat. #:FC-121–2002 | |
| Commercial assay or kit | Nextera XT kit | Illumina | Cat. #:FC-131–1002 | |
| Commercial assay or kit | NEBNext Ultra II DNA Library Prep Kit for Illumina | NEB | Cat. #:E7645 | |
| Software, algorithm | FIJI | PMID:3855844 | | |
| Software, algorithm | SNPsplit | PMID:21493656 | v0.3.2 | |
| Software, algorithm | HISAT2 | PMID:25751142 | v2.0.5 | |
| Software, algorithm | R | R Core Team | 3.5.1 | Available: https://www.R-project.org/ |
| Software, algorithm | featureCounts | PMID:24227677 | | |
| Software, algorithm | Rsubread | PMID:30783653 | 1.32.1 | |
| Software, algorithm | rpart | | R package version 4.1–15 | Available: https://CRAN.R-project.org/package=rpart |
| Software, algorithm | edgeR | PMID:19910308 PMID:22287627 | 3.24.0 | |
| Software, algorithm | Glimma | PMID:28203714 | 1.10.0 | |
| Software, algorithm | ggplot2 | | | Available: https://ggplot2.tidyverse.org |
| Software, algorithm | cowplot | | R package version 1.0.0 | Available: https://CRAN.R-project.org/package=cowplot |
| Software, algorithm | ggbeeswarm | | R package version 0.6.0 | Available: https://CRAN.R-project.org/package=ggbeeswarm |
| Software, algorithm | ggrepel | | R package version 0.8.1 | Available: https://CRAN.R-project.org/package=ggrepel |
| Software, algorithm | ggrastr | | R package version 0.1.7 | |
| Software, algorithm | pheatmap | | R package version 1.0.12 | Available: https://CRAN.R-project.org/package=pheatmap |
| Software, algorithm | TrimGalore! | | v0.4.4 | |

*Continued on next page*

*Continued*

| Reagent type (species) or resource | Designation | Source or reference | Identifiers | Additional information |
|---|---|---|---|---|
| Software, algorithm | Bismark | PMID:21493656 | v0.20.0 | |
| Software, algorithm | FastQC | | v0.11.8 | Available: http://www.bioinformatics.babraham.ac.uk/projects/fastqc |
| Software, algorithm | Bowtie2 | PMID:22388286 | v2.3.4.1 | |
| Software, algorithm | SeqMonk | | v1.45.1 | Available: https://www.bioinformatics.babraham.ac.uk/projects/seqmonk/ |
| Software, algorithm | TMM method | PMID:20196867 | | |
| Software, algorithm | quasi-likelihood F-tests | PMID:27008025 | | |
| Software, algorithm | Benjamini-Hochberg method | 10.1111/j.2517–6161.1995.tb02031.x | | |
| Software, algorithm | Enriched Domain Detector | PMID:24782521 | v1.1.19 | |

## Mouse strains and genotyping

The MMTV-Cre transgene line A (*Wagner et al., 1997*) was backcrossed for more than 10 generations onto the C57BL/6 background from the FVB/N background for use in this study. This was used in combination with a *Smchd1* deleted allele (*Smchd1$^{del}$*) *in trans* to the *Smchd1* floxed (*Smchd1$^{fl}$*) allele (*de Greef et al., 2018*). The *Smchd1$^{del}$* allele was generated from the *Smchd1$^{fl}$* allele using a line of C57BL/6 mice expressing a constitutive Cre transgene (*Orban et al., 1992*). The *Smchd1$^{fl}$* and *Smchd1$^{Del}$* lines were produced and maintained on the C57BL/6 background. The Zp3-Cre line was backcrossed onto the *Smchd1$^{fl}$* allele (*Lewandoski et al., 1997*).

To create the heterozygous placental samples, we used *Smchd1$^{del/fl}$* MMTV-Cre$^{T/+}$ sires mated with Cast/EiJ strain females. The Cast/EiJ strain was purchased from the Jackson laboratory. We observed occasional leakiness of the Cre transgene, which created some mosaic placental samples. These were excluded from our analyses.

To study loss of zygotic SMCHD1 along with loss of oocyte SMCHD1, we produced an F1 line of mice from Cast/EiJ strain dams mated with C57BL/6 *Smchd1$^{del/+}$* sires. By mating with C57BL/6 *Smchd1$^{del/+}$* dams or *Smchd1$^{del/fl}$*; MMTV-Cre$^{T/+}$ dams we could generate embryos where on average half of the imprinted clusters would have a Cast/EiJ allele in trans to C57BL/6, (and therefore be informative for allele-specific analyses) and also be null for Smchd1.

The *Smchd1$^{GFP}$* allele was backcrossed to C57BL/6 for more than 10 generations, and kept as a homozygous breeding line as previously described (*Jansz et al., 2018a*).

Genotyping for the *Smchd1* alleles and sex chromosomes was done as previously described (*Jansz et al., 2018a*). The Cre transgenes were genotyped using a general Cre PCR as previously described (*Leong et al., 2013*).

All post-implantation embryos were generated via natural timed matings. All pre-implantation embryos were generated following superovulation as previously described (*Keniry et al., 2016*).

All illustrations of embryos were adapted from the atlas of embryonic development (*Theiler, 1989*) https://creativecommons.org/licenses/by/3.0/.

## Embryo, placental, and ovary dissections

The embryonic portion of the E14.5 placenta was dissected as previously described (*Mould et al., 2013*). The dissection was learned using a GFP transgene that is transmitted from the sire, and therefore only present in the embryonic portion of the placenta. Based on the structure of the

placenta, the tissue included labyrinth, spongiotrophoblast, and trophoblast giant cells. The yolk sac or a portion of the embryo was taken for genotyping, while the placental piece was snap-frozen for later RNA and DNA preparation. The E9.5 embryos were dissected and snap-frozen whole.

## Sectioning and immunofluorescence studies of the ovary

The ovaries from 6- to 14 week-old mice were harvested and fixed in 10% formalin overnight. Immunofluorescence was performed on paraffin embedded, serially sectioned (4 µm) ovaries. A total of 5–9 slides (three sections per slide) were assessed from each ovary. Briefly, paraffin sections were dewaxed in histolene and heat-induced epitope retrieval was performed in 10 mM sodium citrate buffer (pH 6). Sections were blocked in 10% donkey serum (Sigma Aldrich, D9663) in Tris-sodium chloride (TN) buffer with 3% bovine serum albumin (BSA) (Sigma Aldrich, A9418). Sections were then incubated with buffer only (as negative controls) or primary antibodies for 24 hr at 4°C in TN with 1% BSA in the following dilutions: 1:100 SMCHD1 #5 or #8 (own derivation) and 1:500 CD117/c-kit Antibody (NOVUS af1356). Donkey anti-Goat Alexa 488 (Thermo Fisher Scientific, A-11055) and Donkey anti-Rat IgG DyLight 550 (Invitrogen, SA5-10027) secondary antibodies were applied at 1:500 in TN after washing. Slides were cover-slipped with ProLong Diamond Antifade Mountant with DAPI (Thermo Fisher Scientific, Waltham, MA, P36931) and imaged via a confocal Nikon Eclipse 90i (Nikon Corp., Tokyo, Japan) microscope. Images were processed and analyzed using FIJI software (*Schindelin et al., 2012*). A minimum of 5 follicles for primordial, late secondary follicles and 2–3 antral follicles were tracked and imaged across multiple sections for each genotype.

## Native GFP imaging of pre-implantation embryos

Pre-implantation embryos were collected by flushing oviducts/uteri at 2 cell, 4 cell, 8 cell, 16 cell, 32 cell and blastocyst stages, at E1.5, E2.0, E2.25, E2.75, E3.5 as described previously (*Borensztein et al., 2017*). Embryos collected from $Smchd1^{GFP/GFP}$ females crossed $Smchd1^{GFP/GFP}$ males were used as positive controls and embryos from $Smchd1^{+/+}$ parents were used as negative controls simultaneously with the test embryos to ensure autofluorescence could be accounted for in imaging for GFP. Pre-implantation embryos were fixed with 4% paraformaldehyde for 10 min, washed with PBS, and imaged in Fluorobrite DMEM media (ThermoFisher scientific A1896701) using an AxioObserver microscope (Zeiss) at 20× magnification. Images were processed and analyzed using FIJI software (*Schindelin et al., 2012*). For each experiment, at least three embryos were scored.

## Immunofluorescence of pre-implantation embryos

Embryos were collected from $Smchd1^{GFP/GFP}$ females crossed with $Smchd1^{+/+}$ male and $Smchd1^{+/+}$ females crossed with $Smchd1^{GFP/GFP}$ male as tests, $Smchd1^{+/+}$ crossed with $Smchd1^{+/+}$ as negative controls, and $Smchd1^{GFP/GFP}$ females crossed $Smchd1^{GFP/GFP}$ male as positive controls at 8 cell, 12–16 cell and 24–32 cell stage embryos. The zona pellucida was removed using Acid Tyrode's solution (Sigma T1788), then embryos washed with M2 solution (Sigma M7167) and fixed with 2% PFA for 20 min at room temperature. Embryos were permeabilized in 0.1% Triton-X100 (Sigma T8787) in PBS for 20 min at room temperature. Embryos were blocked in 0.25% gelatin in PBS (gelatin, Sigma, G1393) for 20 min at room temperature. Embryos were transferred to the primary antibody, rabbit anti-GFP (Thermo Fisher A11122, lot 2015993) diluted in block and incubated for 1 hr. Embryos were washed with PBS then transferred to the secondary antibody, anti-rabbit Alexa 488 (Thermofisher A21206, lot 1874771) diluted in block and incubated for 40 min in a dark humidified chamber. The embryos were finally washed with PBS, stained with DAPI for 10 min, washed with PBS again, and mounted into Vectashield H-1000 (Vector labs). The embryos were imaged using the Zeiss LSM 880 system, 40× magnification with airyscan processing. Images were processed and analyzed using FIJI software (*Schindelin et al., 2012*). Negative and positive controls were used to normalize the fluorescence signal.

## Bulk RNA and DNA preparation from embryos and placentae

RNA and DNA were prepared from snap-frozen samples either using a Qiagen All prep kit (Qiagen, Cat # 80204), or a Zymo Quick DNA/RNA miniprep plus kit (Zymo research, Catalog # D7003). DNA and RNA were quantified using nanodrop (Denovix DS-11 spectrophotometer).

## Bulk RNA-sequencing and analysis

Libraries were prepared using TruSeq V1 or V2 RNA sample preparation kits from 500 ng total RNA as per manufacturers' instructions. Fragments above 200 bp were size-selected and cleaned up using AMPure XP magnetic beads. Final cDNA libraries were quantified using D1000 tape on the TapeStation (4200, Agilent Technologies) for sequencing on the Illumina Nextseq500 platform using 80 bp, paired-end reads.

RNA-seq reads were trimmed for adapter and low quality sequences using TrimGalore! v0.4.4, before mapping onto the GRCm38 mouse genome reference N-masked for Cast SNPs prepared with SNPsplit v0.3.2 (*Krueger and Andrews, 2016*) with HISAT2 v2.0.5 (*Kim et al., 2015*), in paired-end mode and disabling soft-clipping. Alignments specific to the C57BL/6 and Cast alleles were separated using SNPsplit v0.3.2 in paired-end mode.

Gene counts were obtained in R 3.5.1 (*R Development Core Team, 2019*) from the split and non-split bam files with the featureCounts function from the Rsubread package (1.32.1 *Liao et al., 2014*; *Liao et al., 2019*), provided with the GRCm38.90 GTF annotation downloaded from Ensembl, and ignoring multi-mapping or multi-overlapping reads.

Because the samples for the maternal and zygotic deletion experiment are first-generation backcrosses of CastB6F1s to C57BL/6, differential expression between alleles can only be performed in genomic regions that are heterozygous (half of the genome, on average). To identify these regions for each sample, we fit a recursive partition tree with R rpart (*R Development Core Team, 2019*; *Therneau and Atkinson, 2019*; *Breiman et al., 1984*) function to the proportion of Cast/EiJ reads in each 100 kb bin tiling the genome, with options minsplit = 4, cp = 0.05.

Genes were defined as expressed and retained for differential expression analysis if they had at least one count per million (cpm) in at least a third of the libraries in the non-haplotyped data.

For analysis of global expression changes, total (non-haplotyped) counts were normalized in edgeR v3.24.0 (*Robinson et al., 2010*; *McCarthy et al., 2012*) with the TMM method (*Robinson and Oshlack, 2010*), and differential expression analysis performed with quasi-likelihood F-tests (*Lun et al., 2016*). P-values were corrected with the Benjamini-Hochberg method (*Benjamini and Hochberg, 1995*). Differential expression results were visualized with Glimma 1.10.0 (*Su et al., 2017*), with differential expression cut-offs of adjusted p-value<0.05 and log2-fold-change>1. Plots were generated in R with the ggplot2 (*Wickham, 2016*), cowplot (*Wilke, 2019*), ggbeeswarm (*Clarke and Sherrill-Mix, 2017*), ggrepel (*Slowikowski, 2019*), ggrastr (*Petukhov, 2019*), and pheatmap (*Kolde, 2019*) packages.

Testing for changes in imprinted expression was performed as follows: we compiled a list of all 316 known mouse imprinted genes and kept autosomal genes expressed in the experiment of interest (based on total counts, as above). *Gatm* was removed because the biased maternal expression is a result of maternal decidua contamination in the embryonic placenta dissection (*Okae et al., 2012*). To determine whether they were imprinted in the tissue of interest, we then fitted a logistic regression with the glm function from the stats package (*R Development Core Team, 2019*) on the maternal and paternal counts for each gene in the wild-type samples and retained the genes with an absolute log-ratio of expression greater than or equal to log(1.5). Genes without on average at least 10 haplotyped counts per sample for each genotype were also excluded from further analysis.

To investigate the effect of *Smchd1* oocyte-deletion on imprinted expression, we used edgeR's paired-sample design (*Chen et al., 2017*) with tagwise dispersion to model the maternal and paternal counts as a function of genotype (and Cre-deletion, in the case of placental samples where we had both Zp3-Cre and MMTV-Cre samples). We tested the effect of genotype with a likelihood ratio test. The p-values were corrected with the Benjamini-Hochberg method. Genes were considered differentially imprinted when the adjusted p-value was less than 0.05 and the absolute difference in silent allele proportion average between wild-type and oocyte-deleted samples was greater than 5% (for at least one Cre construct in the case of the placenta).

We could not use the same approach for the zygotic-and-oocyte deletion experiment as the genetic heterogeneity of the samples resulted in a haplotyped counts matrix with about half of the values missing. Instead, we fitted a beta-binomial regression with the betabin function from package the aod (*Lesnoff and Lancelot, 2012*) package on the maternal and paternal counts for each gene with informative samples, with a fixed dispersion of 0.01 estimated from the oocyte-only deletion

data. We used the same criteria of 5% false discovery (FDR) and 5% absolute difference in silent allele proportion to call differential imprinting.

## Bulk reduced representation bisulfite sequencing and analysis

DNA extracted above was cleaned up using Zymo research DNA Clean and concentrator-5 kit. DNA was quantified using the Qubit dsDNA assay kit (Thermo Fisher Scientific Q32853) and 100 ng were used as input for library preparation with the NuGEN Ovation RRBS methyl-seq system (Integrated sciences). Bisulfite conversion was carried out using QIAGEN EpiTect Fast DNA Bisulfite Kit. Quantitative and Qualitative analysis of library prep was carried out using D1000 tape on TapeStation 2200 (Agilent Technologies). Samples were sequenced on the HiSeq2500 platform using 100 bp paired-end reads.

Paired RRBS reads were trimmed first for an adapter and low-quality sequences with TrimGalore! v0.4.4 specifying the options -a AGATCGGAAGAGC -a2 AAATCAAAAAAAC, and then for the diversity bases introduced during library preparation with the trimRRBSdiversityAdaptCustomers.py script provided by NuGEN (*Lovci, 2018*). Trimmed reads were mapped onto the GRCm38 mouse genome reference N-masked for Cast SNPs with Bismark v0.20.0 (86) and the alignments were split by allele with SNPsplit v0.3.2 in bisulfite mode. Methylation calls were extracted with Bismark's bismark_methylation_extractor function.

We compiled a list of 45 known imprinted Differentially Methylated Regions (DMRs) and aggregated the haplotyped methylated and unmethylated counts over these regions. Regions with fewer than 10 counts per sample on average for wild-type and oocyte-deleted genotypes were filtered out, as were regions with an average methylation difference between the parental alleles in wild-type samples of less than 10%. Methylated and unmethylated counts for the hypermethylated allele were analyzed with edgeR's paired-sample design, setting the dispersion as the common dispersion (*Chen et al., 2017*). The p-values were adjusted for multiple testing with the Benjamini-Hochberg correction. Hypermethylated alleles were considered to be differentially methylated between wild-type and oocyte-deleted samples when the adjusted FDR was below 0.05.

To analyze methylation on CpG islands across the genome, we collected total (non-split by allele) methylated and unmethylated counts over the mm10 CpG island annotation from Seqmonk \cite (*Andrews, 2007*). Regions with fewer than 10 counts were excluded, and we subjected the remaining loci to a logistic regression followed by Benjamini-Hochberg correction of the p-values. We considered CpG islands as differentially methylated when the average difference in methylation proportion between wild-type and maternally-deleted samples was at least 10% and the adjusted p-value was below 0.05. We associated CpG islands with genes by annotating the island with its closest gene within 2 kb.

## Smchd1-GFP ChIP-Seq in neural stem cells

Three primary female $Smchd1^{GFP/GFP}$ NSC lines were derived as previously described (*Chen et al., 2015*) and harvested with Accutase (Sigma-Aldrich), washed with culture medium, and PBS and cross-linked with 1% formaldehyde (vol/vol) for 10 min at room temperature with rotation, and subsequently quenched with glycine. The cells were immediately pelleted at 456 $g$ for 5 min at 4°C and washed twice in cold PBS. The crosslinked pellets were then snap-frozen in dry ice.

For each cell line, $4 \times 10^7$ crosslinked nuclei were extracted from frozen pellets by incubating on ice for 10 min in 14 mL of ChIP buffer (150 mM NaCl, 50 mM) Tris-HCl pH 7.5, 5 mM EDTA, 0.5% vol/vol Igepal CA-630, 1% Triton X-100, 1× cOmplete cocktail (Roche), and homogenizing 25 times in a tight dounce on ice. Nuclei were pelleted at 12,000 $g$ for 1 min at 4°C, washed with ChIP buffer then resuspended in 1.6 mL of MNase buffer with 1× BSA (NEB). The nuclei solution was preincubated at 37°C for 5 min, then $2 \times 10^4$ U of MNase (NEB) was added and incubated for a further 5 min. The reaction was stopped by adding 10 mM of EDTA and incubating on ice for 10 min. Nuclei were pelleted (4°C, 12,000 $g$, 1 min), resuspended in 520 µL of ChIP buffer then sonicated with a Covaris S220 sonicator (peak power, 125; duty factor, 10; cycle/burst, 200; duration, 15 s) in Covaris microTubes. Sonicated solution was diluted 10 times with ChIP buffer then spun at 12,000 $g$ at 4°C for 1 min to clear debris. 20 uL of the supernatant was taken for the whole-cell extract (WCE) samples and the rest was used for immunoprecipitation overnight at 4°C with rotation with 16 µg of anti-GFP antibody (Invitrogen A11122). The chromatin was then cleared by centrifugation (12,000 $g$ for

10 min at 4°C) and 80 µL of protein G DynaBeads (Thermo Fisher, washed three times in cold ChIP buffer right before use) were added before incubating at 4°C for 1 hr with rotation. The samples were then washed six times with cold ChIP buffer. The antibody-bound chromatin was eluted from the beads with two rounds of 400 µL of elution buffer (1% SDS, 0.1 M NaHCO$_3$) by rotating at room temperature for 15 min. 8 µL of 5M NaCl and 1 µL of RNase A (NEB) were added to every 200 µL of eluate and to the WCE samples (diluted to 200 µL with elution buffer), incubated overnight at 65°C to reverse crosslinking then treated with 4 µL of 20 µg/mL Proteinase K at 65°C for 1 hr. DNA was extracted with Zymo ChIP DNA clean and concentrator kit.

Libraries were generated with an Illumina TruSeq DNA Sample Preparation Kit. 200- to 400 bp fragments were size-selected with AMPure XP magnetic beads. Libraries were quantified with a D1000 tape in a 4200 Tapestation (Agilent). Libraries were pooled and sequenced on the Illumina NextSeq platform, with 75 bp single-end reads.

Adapter trimming was performed with Trim galore! v0.4.4, library QC was assessed with FastQC v0.11.8 (*Andrews, 2010*) before mapping with Bowtie2 v2.3.4.1 (*Langmead and Salzberg, 2012*) with default options to the GRCm38.p6 version of the reference mouse genome. BAM files were imported into SeqMonk v1.45.1 (*Andrews, 2007*) extending them by 150 bp and peaks were called with the MACS-style caller within the SeqMonk package (settings for 300 bp fragments, p<1 × 10$^{-5}$) by merging all three Smchd1-GFP IP and both input biological replicates into replicate sets. Larger regions enriched for Smchd1 binding were called with Enriched Domain Detector (https://academic. oup.com/nar/article/42/11/e92/1432049) with a 3 kb bin size.

## SMCHD1-GFP ChIP-Seq in E14.5 placenta

The ChIP protocol was adapted from *Skene and Henikoff, 2017*. Three snap-frozen E14.5 placentae (about 150 mg each) from a B6Cast F1 mouse with a heterozygous SMCHD1-GFP tag were ground in liquid nitrogen with a mortar and pestle. The tissue powder was resuspended in ice-cold PBS with proteinase inhibitors and spun for 3 min at 600 g, at 4°C. The pellet was resuspended in 1 mL PBS at room temperature for a 15 min fixation with 1% formaldehyde. After quenching for 2 min with 125 mM glycine, the cross-linked tissue was pelleted as before and washed twice with ice-cold PBS with proteinase inhibitors. Nuclei were extracted by incubating in 600 uL ice-cold lysis buffer for 10 min. After dilution with 5.4 mL ChIP dilution buffer and pre-warming at 37°C, 15 uL MNase (NEB, 2000 U/ uL) were added for a 15 min digestion at 37°C. The digestion was stopped with 10 mM EDTA and 20 mM EGTA, and the nuclei were pelleted by centrifugation at 12,000 g for 1 min at 4°C. The rest of the steps followed the above protocol for ChIP-seq in NSCs.

Libraries were made from the immunoprecipitated DNA and the input control using the NEB Ultra II DNA library preparation kit, using 1:10 adaptor dilution and 11 cycles of PCR. Libraries were sequenced in 80PE on a NextSeq.

Adapter trimming was performed with Trim galore! v0.4.4, library QC was assessed with FastQC v0.11.8 (*Clarke and Sherrill-Mix, 2017*) before mapping with Bowtie2 v2.3.4.1 (*Slowikowski, 2019*) in paired-end mode with default options to the GRCm38.p6 version of the reference mouse genome with N-masked Cast SNPs.

Regions enriched for SMCHD1 binding were called with Enriched Domain Detector (https://academic.oup.com/nar/article/42/11/e92/1432049) with a 3 kb bin size and peaks were called with MACS2 with the 'broad peaks' setting (*Zhang et al., 2008*).

## Single pre-implantation embryo M and T-seq

E2.75 embryos were sorted into a preprepared lysis buffer containing 2.5 µL of RLT Plus (Qiagen) with 1 U/µL SUPERase-IN (Ambion). Genomic DNA and mRNA were separated using oligo-dT conjugated magnetic beads according to the G and T-seq protocol (*Macaulay et al., 2015*), however, gDNA was eluted into 10 µL of H$_2$O and 16 cycles of amplification followed for cDNA synthesis.

Bisulfite libraries were prepared using an adapted scBS-seq protocol (*Angermueller et al., 2016*). Briefly, bisulfite conversion was performed before introducing Illumina adaptor sequences with random priming oligos. Only one round of first and second strand synthesis was performed, using primers compatible with the NEBNext multiplex oligos for library production (Forward: 5'-C TACACGACGCTCTTCCGATCTNNNNNNN-3';   Reverse:   5'-   CAGACGTGTGCTCTTCCGATC TNNNNNN-3'). Sixteen cycles of library amplification followed and all Ampure XP bead (Beckman

Coulter) purifications were performed at 0.65×. To assess the library quality and quantity a High-Sensitivity D5000 ScreenTape on the Agilent TapeStation and the ProNex NGS Library Quantification Kit were used. Single-embryo bisulfite libraries were pooled for 150 bp paired-end sequencing on a NovaSeq6000.

RNA-seq libraries were prepared from amplified cDNA using the Nextera XT kit (Illumina) according to the manufacturer's guidelines but using one-fifth volumes. Single-embryo RNA-seq libraries were pooled for 75 bp paired-end sequencing on a NextSeq500.

## Single pre-implantation embryo M and T-seq analysis

### Methylome analysis
To remove sequence biases due to the post-bisulfite adapter tagging, the first nine base pairs from read 1 and read two were removed with Trim Galore! v0.4.4 (http://www.bioinformatics.babraham.ac.uk/projects/trim_galore/), in addition to adapter and quality trimming, with options (–`clip_R1` 9 –`clip_R2` 9). The reads were then first mapped to the Cast-masked GRCm38 genome in paired-end mode with Bismark v0.20.0 (*Krueger and Andrews, 2011*) with the pbat option to allow dovetail mapping, and keeping unmapped reads (–`unmapped`). Unmapped reads 1 and 2 were then mapped in single-end mode, with the –pbat option set for read 1. Each alignment file (paired, unmapped reads 1, and unmapped reads 2) was then deduplicated with deduplicate_bismark, and haplotyped with SNPsplit v0.3.2. Methylation calls were obtained with bismark_methylation_extractor and pooled (paired, unmapped reads one and unmapped reads 2).

Embryos were sexed based on their Cast-haplotyped bisulfite reads mapping to the X chromosome. Embryos that did not produce a library, showed evidence of maternal contamination (much higher proportion of C57BL/6 reads compared to Cast reads in the methylome and transcriptome data), or displayed chromosomal abnormalities were excluded.

Differences in methylation at known imprinted DMRs were tested in edgeR as for the bulk samples, although we also tested the non-split data in addition to the hypermethylated allele to increase the coverage and have more testable regions.

The whole-genome differential methylation analysis of the non-split data was performed on CpG islands, on gene transcriptional start sites (−4 kb to +1 kb of the TSS) and over non-overlapping 10 kb windows tiling the genome, testing methylation counts in edgeR as before.

## Transcriptome analysis
RNA-seq analyses for the single embryos passing quality control were performed as for the bulk samples, except that the dispersion was set as the common dispersion (rather than tagwise) to avoid unreliable estimates of dispersion for individual genes on sparser single-embryo expression data.

## Acknowledgements
We thank Kay-Uwe Wagner, Jane Visvader, Patrick Western, and Graham Kay for provision of the MMTV-Cre, Zp3-Cre, and Smchd1-GFP mice used in the study. We thank Stephen Wilcox and Jafar Jabbari for sequencing. We thank Peter Hickey, Yunshun Chen, and Gordon Smyth for statistical advice and Natasha Jansz for critical reading of the manuscript.

## Additional information

### Funding

| Funder | Grant reference number | Author |
|---|---|---|
| National Health and Medical Research Council | 1098290 | Matthew E Ritchie Marnie E Blewitt |
| Bellberry-Viertel Senior Medical Research Fellowship | | Marnie E Blewitt |

The funders had no role in study design, data collection and interpretation, or the decision to submit the work for publication.

## Author contributions
Iromi Wanigasuriya, Conceptualization, Data curation, Formal analysis, Methodology, Writing - original draft, Writing - review and editing; Quentin Gouil, Conceptualization, Data curation, Software, Formal analysis, Investigation, Methodology, Writing - original draft, Writing - review and editing; Sarah A Kinkel, Conceptualization, Formal analysis, Investigation, Writing - review and editing; Andrés Tapia del Fierro, Data curation, Formal analysis, Investigation, Methodology, Writing - review and editing; Tamara Beck, Ellise A Roper, Kelsey Breslin, Investigation, Methodology; Jessica Stringer, Data curation, Investigation, Methodology; Karla Hutt, Data curation, Investigation, Writing - review and editing; Heather J Lee, Conceptualization, Resources, Investigation, Methodology, Writing - review and editing; Andrew Keniry, Conceptualization, Data curation, Formal analysis, Supervision, Writing - review and editing; Matthew E Ritchie, Resources, Data curation, Formal analysis, Supervision, Funding acquisition, Methodology, Writing - review and editing; Marnie E Blewitt, Conceptualization, Resources, Supervision, Funding acquisition, Writing - original draft, Writing - review and editing

## Author ORCIDs
Iromi Wanigasuriya (iD) https://orcid.org/0000-0002-2941-8764
Quentin Gouil (iD) https://orcid.org/0000-0002-5142-7886
Karla Hutt (iD) http://orcid.org/0000-0002-5111-8389
Marnie E Blewitt (iD) https://orcid.org/0000-0002-2984-1474

## Ethics
Animal experimentation: This study was performed in accordance with the recommendations of the Australian code for ethical use of laboratory animals. All animals were handled according to approved institutional animal ethics committee protocols (WEHI AEC 2014.026, 2018.004).

## Decision letter and Author response
Decision letter https://doi.org/10.7554/eLife.55529.sa1
Author response https://doi.org/10.7554/eLife.55529.sa2

# Additional files

## Supplementary files
• Supplementary file 1. All tables of statistical data for allele-specific RNA-seq analyses at imprinted genes. (a) Differential imprinted expression in male E14.5 B6Cast placentae, Smchd1 matΔ versus wt. A gene is called differentially imprinted if the multiple-testing corrected p-value is below 0.05 and the absolute difference in silent allele proportion greater than 0.05 (b) Differential imprinted expression in male E14.5 CastB6 placentae, Smchd1 het versus wt. A gene is called differentially imprinted if the multiple-testing corrected p-value is below 0.05 and the absolute difference in silent allele proportion greater than 0.05 (c) Differential imprinted expression in male E9.5 B6Cast embryos, Smchd1 matΔ versus wt. A gene is called differentially imprinted if the multiple-testing corrected p-value is below 0.05 and the absolute difference in silent allele proportion greater than 0.05. Gnas is likely a false positive because in the second dataset of E9.5 embryos (maternal and zygotic deletions) it did not display imprinted expression in the wt samples (d) Differential imprinted expression in male E2.75 B6Cast embryos, Smchd1 matΔ versus wt. A gene is called differentially imprinted if the multiple-testing corrected p-value is below 0.05 and the absolute difference in silent allele proportion greater than 0.05 (e) Differential imprinted expression in male E14.5 B6(CastB6) placentae, Smchd1 matΔ versus wt. A gene is called differentially imprinted if the multiple-testing corrected p-value is below 0.05 and the absolute difference in silent allele proportion greater than 0.05 (f) Differential imprinted expression in male E14.5 B6(CastB6) placentae, Smchd1 zygΔ versus wt. A gene is called differentially imprinted if the multiple-testing corrected p-value is below 0.05 and the absolute difference in silent allele proportion greater than 0.05 (g) Differential imprinted expression in male E14.5 B6(CastB6) placentae, Smchd1 matzygΔ versus wt. A gene is called differentially

imprinted if the multiple-testing corrected p-value is below 0.05 and the absolute difference in silent allele proportion greater than 0.05 (h) Differential imprinted expression in male E14.5 B6(CastB6) placentae, Smchd1 zygΔ versus matΔ. A gene is called differentially imprinted if the multiple-testing corrected p-value is below 0.05 and the absolute difference in silent allele proportion greater than 0.05 (i) Differential imprinted expression in male E14.5 B6(CastB6) placentae, Smchd1 matzygΔ versus matΔ. A gene is called differentially imprinted if the multiple-testing corrected p-value is below 0.05 and the absolute difference in silent allele proportion greater than 0.05 (j) Differential imprinted expression in male E14.5 B6(CastB6) placentae, Smchd1 matzygΔ versus zygΔ. A gene is called differentially imprinted if the multiple-testing corrected p-value is below 0.05 and the absolute difference in silent allele proportion greater than 0.05 (k) Differential imprinted expression in male E9.5 B6(CastB6) embryos, Smchd1 matΔ versus wt. A gene is called differentially imprinted if the multiple-testing corrected p-value is below 0.05 and the absolute difference in silent allele proportion greater than 0.05 (l) Differential imprinted expression in male E9.5 B6(CastB6) embryos, Smchd1 zygΔ versus wt. A gene is called differentially imprinted if the multiple-testing corrected p-value is below 0.05 and the absolute difference in silent allele proportion greater than 0.05 (m) Differential imprinted expression in male E9.5 B6(CastB6) embryos, Smchd1 matzygΔ versus wt. A gene is called differentially imprinted if the multiple-testing corrected p-value is below 0.05 and the absolute difference in silent allele proportion greater than 0.05 (n) Differential imprinted expression in male E9.5 B6(CastB6) embryos, Smchd1 zygΔ versus matΔ. A gene is called differentiallly imprinted if the multiple-testing corrected p-value is below 0.05 and the absolute difference in silent allele proportion greater than 0.05 (o) Differential imprinted expression in male E9.5 B6(CastB6) embryos, Smchd1 matzygΔ versus matΔ. A gene is called differentiallly imprinted if the multiple-testing corrected p-value is below 0.05 and the absolute difference in silent allele proportion greater than 0.05 (p) Differential imprinted expression in male E9.5 B6(CastB6) embryos, Smchd1 matzygΔ versus zygΔ. A gene is called differentiallly imprinted if the multiple-testing corrected p-value is below 0.05 and the absolute difference in silent allele proportion greater than 0.05.

• Supplementary file 2. Tables of statistical data for whole-genome RNA-seq analyses. (a) Differential expression (non-split counts) in male E9.5 B6Cast embryos, Smchd1 matΔ versus wt. A gene is called differentially expressed if the q-value is below 0.05 and the fold change is greater than 2. (b) Differential expression (non-split counts) in male E14.5 B6Cast embryonic placentae, Smchd1 matΔ versus wt (MMTV-Cre). A gene is called differentially expressed if the q-value is below 0.05 and the fold change is greater than 2. (c) Differential expression (non-split counts) in male E14.5 B6Cast embryonic placentae, Smchd1 matΔ versus wt (Zp3-Cre). A gene is called differentially expressed if the q-value is below 0.05 and the fold change is greater than 2. (d) Differential expression (non-split counts) in male E14.5 CastB6 embryonic placentae, Smchd1 het versus wt. A gene is called differentially expressed if the q-value is below 0.05 and the fold change is greater than 2. (e) Differential expression (non-split counts) in male E2.75 B6Cast embryos, Smchd1 matΔ versus wt. A gene is called differentially expressed if the q-value is below 0.05 and the fold change is greater than 2.

• Supplementary file 3. Tables of statistical data for allele-specific DNA methylation analyses at imprinted DMRs. (a) Differential imprinted methylation in E14.5 B6Cast male placentae, Smchd1 matΔ versus wt. A methylated allele is called differentially methylated if the multiple-testing corrected p-value is below 0.05. (b) Differential imprinted methylation in E2.75 B6Cast male embryos, Smchd1 matΔ versus wt. A methylated allele is called differentially methylated if the multiple-testing corrected p-value is below 0.05. (c) Differential imprinted methylation in E2.75 B6Cast male embryos, Smchd1 matΔ versus wt. A methylated allele is called differentially methylated if the multiple-testing corrected p-value is below 0.05.

• Supplementary file 4. Tables of statistical data for genome-wide DNA methylation analyses. (a) Differentially methylated CGIs in E14.5 B6Cast male placentae, Smchd1 matΔ versus wt. A CGI is called differentially methylated if the multiple-testing corrected p-value is below 0.05 and the absolute difference in methylation level between the two conditions is at least 10%. (b) Differentially methylated CGIs in E2.75 B6Cast male embryos, Smchd1 matΔ versus wt. A CGI is called differentially methylated if the multiple-testing corrected p-value is below 0.05 and the absolute difference in methylation level between the two conditions is at least 10%. (c) Differentially methylated promoters (−3 to +1 kb) in E2.75 B6Cast male embryos, Smchd1 matΔ versus wt. A CGI is called differentially

methylated if the multiple-testing corrected p-value is below 0.05 and the absolute difference in methylation level between the two conditions is at least 10%. (d) Differentially methylated bins (10 kb, tiling the genome without overlaps) in E2.75 B6Cast male embryos, Smchd1 matΔ versus wt. A CGI is called differentially methylated if the multiple-testing corrected p-value is below 0.05 and the absolute difference in methylation level between the two conditions is at least 10%.

- Transparent reporting form

## Data availability

All the sequencing data for this study are available from the Short Read Archive under the BioProject accession PRJNA530651. Results of the genome-wide expression analyses can be consulted on http://bioinf.wehi.edu.au/smchd1_mat_effects/.

The following dataset was generated:

| Author(s) | Year | Dataset title | Dataset URL | Database and Identifier |
|---|---|---|---|---|
| Wanigasuriya I, Gouil QA, Kinkel SA, Fierro AT, Beck T, Roper EA, Breslin K, Lee HJ, Keniry A, Ritchie ME, Blewitt ME | 2019 | Smchd1 is a maternal effect gene required for autosomal imprinting | https://www.ncbi.nlm.nih.gov/sra/?term=PRJNA530651 | NCBI Sequence Read Archive, PRJNA530651 |

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
