## [Decision Letter]

**Acceptance summary:**

This study highlights the divergent mechanisms that control the allelic expression of canonical (DNA methylation-dependent) imprinted genes versus non-canonical (polycomb-dependent) imprinted genes. This has implications in epigenetics but also for the etiology of syndromes related to mutations in SMCHD1.

**Decision letter after peer review:**

Thank you for submitting your article "*Smchd1* is a maternal effect gene required for autosomal imprinting" for consideration by *eLife*. Your article has been reviewed by three peer reviewers, including Deborah Bourc’his as the Reviewing Editor and Reviewer #1, and the evaluation has been overseen by Kevin Struhl as the Senior Editor. The following individual involved in review of your submission has agreed to reveal their identity: Adele Murrell (Reviewer #3).

The reviewers have discussed the reviews with one another and the Reviewing Editor has drafted this decision to help you prepare a revised submission.

Summary:

Imprinted gene expression relies on parent-specific epigenetic marks that are laid down in the parental gametes and maintained after fertilization. DNA methylation was thought to be the only mark to control imprinting, but oocyte H3K27me3 was recently shown to cross fertilization, to be maintained for a few cell cycles in the embryo, and to trigger the regulation of (a few) imprinted genes in the placenta after implantation. The authors report here the involvement of the chromatin SCMCHD1 protein as a maternal regulator of a defined set of imprinted genes, using allelic RNA-seq, DNA methylation by RRBS, single embryo RNA-seq and methylome analyses. The authors observed partial reactivation of the normally silent allele of some imprinted genes in the placenta of maternally SMCHD1-depleted conceptuses. This effect is independent of DNA methylation and seems to affect specifically genes that are regulated by H3K27me3, either maternally provided or established in the early embryo, independently from their expression in the early embryo.

The authors propose that maternal SMCHD1 may consolidate the chromatin state that is inherited by some imprinted genes in the early embryo, and this may be required for the "long-lived" epigenetic regulation of allele-specific repression. In contrast to previously known maternal regulators of imprinting, the effect is independent from DNA methylation. While the data are sound and the manuscript well written, the study is quite succinct, very descriptive and with limited breadth. Notably, genome-wide assays are not fully exploited, it seems both biased and superficial to narrow the analysis to imprinted genes only. The authors could likely learn more about the role of maternal SMCHD1 by looking at non-imprinted targets.

Essential revisions:

1) Some canonical imprinted genes that show effects in the current analysis have been originally identified as maternally expressed in the placenta only. The authors should be aware that the imprinting status of these genes can be fallacious, resulting from contamination with maternal material (See Proudhon and Bourc'his, 2010, Okae et al., 2012). On this matter, *Gatm* is no longer considered as a truly maternally expressed imprinted gene, but rather as a gene that is very highly expressed in maternal decidua cells, which cannot be totally removed during placenta isolations (Oake et al., 2012). This is actually a very good indicator of the level of maternal contamination in placenta dissections, and from the heatmap in Figure 2—figure supplement 1A, two samples appear with particularly high degree of maternal contamination. Please remove the inclusion of *Gatm* from the manuscript.

2) Considering the recognized role of SMCHD1 as a transcriptional repressor, could the authors specify what are the genes that are upregulated in the RNA-seq of Zp3-CRE (n=11) and MMTV-CRE (n=-3) *Smchd1*-deleted placentae? And what is the unique gene that is in common between the two models? Are these genes also misregulated in the E9.5 embryo sample (MMTV-CRE )? Do they show relationships with H3K27me3-centered regulation?

3) Regarding the DNA methylation analyses of maternal SMCHD1 deficiency (RRBS and single embryos): although the authors could not see DNA methylation defects at imprinted loci, what about the rest of the genome? Was there any differentially methylated regions elsewhere?

4) Because the authors do not mention it, I suspect that imprinted genes were not among the upregulated genes of the RNA-seq. Could the authors clearly state this in the text? Although the authors report the relative expression of parental alleles, they never provide quantitative values of global expression of these genes showing allelic disturbance. Please provide quantitative RT-PCR of the expression levels of imprinted genes that show the highest degree of reactivation of the normally silent allele in placentae.

5) The authors fail to mention in the text and the figure legends from which ChIP datasets the SMCHD1 binding sites (reported in Figures 2, and 3) are extracted. However, from the Materials and methods section and the legend of Supplementary Figure 3B, it appears that the ChIP-seq was performed in neural stem cells. What is the relevance of reporting the binding sites that SMCHD1 has in neurons relative to expression patterns that were measured in placentae? ChIP-seq analysis in E14.5 placenta (from Smchd1-GFP mouse) is needed.

6) The emphasis on the "long-lived" epigenetic effects of SMCHD1 presence in the embryo is overrated. First, effects are mostly seen in the placenta, a short-lived organ whose existence is limited to gestation. Then, we would like to see some late placenta to see whether the effects are maintained, and earlier placenta, to see whether the effects may be reducing with time. Targeted approaches could be used here. If you do not provide these data, please refrain from using "long-lived".

7) The lack of apparent phenotypic effects in the SMCHD1 maternally deficient progenies (at least until weaning age) suggests that the biological relevance of such regulation is weak. This should be clearly acknowledged in the Discussion. Moreover, if SMCHD1 effects are linked to H3K27me3, is there some phenotypic overlap with mouse models of maternal H3K27me3 deficiency (Prokopuk et al., 2018)? In other words, although litter size is normal at weaning, what about the weight? Any noticeable difference at birth or any time before weaning?

8) Please state that other non-canonical imprinted genes were not affected by Smchd1 maternal effect. If the authors do not state it clearly, readers have to perform searches of the supplementary data.

9) Genes in the chr7 imprinting cluster (*Tssc4* and *Ascl2*) show clear reactivation (Figure 3). Seeing that imprinting of the neighbouring *Cdkn1c* and *Phlda2* also depends on allelic repressive histone modifications, the authors should report if these are susceptible to LOI and if there are any changes to the *Cdkn1c* somatic DMR.

10) Please state why E14.5 placentae were analyzed yet E9.5 embryos? Why not at the same developmental age?

11) On the same matter, please state what cell types were collected and analyzed when referring to "embryonic proportion of the placenta" (Labyrinth, Spongiotrophoblast, giant trophoblast etc?).

---

## [Author Response]

Summary:Imprinted gene expression relies on parent-specific epigenetic marks that are laid down in the parental gametes and maintained after fertilization. DNA methylation was thought to be the only mark to control imprinting, but oocyte H3K27me3 was recently shown to cross fertilization, to be maintained for a few cell cycles in the embryo, and to trigger the regulation of (a few) imprinted genes in the placenta after implantation. The authors report here the involvement of the chromatin SCMCHD1 protein as a maternal regulator of a defined set of imprinted genes, using allelic RNA-seq, DNA methylation by RRBS, single embryo RNA-seq and methylome analyses. The authors observed partial reactivation of the normally silent allele of some imprinted genes in the placenta of maternally SMCHD1-depleted conceptuses. This effect is independent of DNA methylation and seems to affect specifically genes that are regulated by H3K27me3, either maternally provided or established in the early embryo, independently from their expression in the early embryo.The authors propose that maternal SMCHD1 may consolidate the chromatin state that is inherited by some imprinted genes in the early embryo, and this may be required for the "long-lived" epigenetic regulation of allele-specific repression. In contrast to previously known maternal regulators of imprinting, the effect is independent from DNA methylation. While the data are sound and the manuscript well written, the study is quite succinct, very descriptive and with limited breadth. Notably, genome-wide assays are not fully exploited, it seems both biased and superficial to narrow the analysis to imprinted genes only. The authors could likely learn more about the role of maternal SMCHD1 by looking at non-imprinted targets.

We now fully describe the results of the genomic data, explaining that we concentrated on the role of maternal Smchd1 at imprinted genes because the maternal deletion has limited impact on genome-wide gene expression and DNA methylation.

We also performed Smchd1 ChIP in placenta and analysed these data in relation to histone marks including H3K27me3 to strengthen the manuscript as requested by the reviewers.

Essential revisions:1) Some canonical imprinted genes that show effects in the current analysis have been originally identified as maternally expressed in the placenta only. The authors should be aware that the imprinting status of these genes can be fallacious, resulting from contamination with maternal material (See Proudhon and Bourc'his, 2010, Okae et al., 2012). On this matter, Gatm is no longer considered as a truly maternally expressed imprinted gene, but rather as a gene that is very highly expressed in maternal decidua cells, which cannot be totally removed during placenta isolations (Oake et al., 2012). This is actually a very good indicator of the level of maternal contamination in placenta dissections, and from the heatmap in Figure 2—figure supplement 1A, two samples appear with particularly high degree of maternal contamination. Please remove the inclusion of Gatm from the manuscript.

Thank you for highlighting these papers for us. We have removed *Gatm* from the revised manuscript.

2) Considering the recognized role of SMCHD1 as a transcriptional repressor, could the authors specify what are the genes that are upregulated in the RNA-seq of Zp3-CRE (n=11) and MMTV-CRE (n=-3) Smchd1-deleted placentae? And what is the unique gene that is in common between the two models? Are these genes also misregulated in the E9.5 embryo sample (MMTV-CRE )? Do they show relationships with H3K27me3-centered regulation?

We appreciate that more fully detailing the differential expression genome-wide is useful for readers, and may reveal a maternal effect of Smchd1 at non-imprinted targets as well. The full results of the differential expression analyses of total counts are now provided in Supplementary file 2 (results of the allelic proportion analyses are in Supplementary file 1) and we explicitly compared the results from the different tissues and Cre models (Figure 1—figure supplement 2A). The text now reads:

“We found only five consistently differentially expressed genes between *Smchd1*^mat∆^

and control placental samples from both Cre models, with modest fold-changes (Figure 1—figure supplement 2A-C, Supplementary files 1-2): one upregulated (*Gm8493*) and four downregulated (*Ceacam12*, *Psg16*, *Gm7863*and *Afp*). […] These data are consistent with no observable effect of deleting maternal Smchd1 on viability or postnatal weight (Figure 1—figure supplement 1A and B).”

Because of the modest fold-changes, limited overlap between the two Cre experiments in placenta and the two tissues (see Figure 1—figure supplement 2A), in addition to potential for litter effects as the matΔ and wt samples cannot be litter-matched, we did not want to overinterpret the larger DE in E9.5 embryo data. The tighter MA-plot from the litter-matched het and wt placenta samples (Figure 2—figure supplement 2B) compared to MMTV-Cre mat placenta is consistent with litter-to-litter variability being the potential cause of the larger DE in E9.5 embryos. In the E9.5 embryo data, other clustered gene families such as cathepsins on chromosome 13 (*Cts3, Cts6, Ctsj, Ctsq*), prolactin genes in a different region of chromosome 13 (*Prl7a2, Prl7d1, Prl8a8, Prl8a9*), or carcino-embryonic antigen-like cellular adhesion molecules (CEACAM) genes on chromosome 7 were also significantly upregulated, but this was driven by one maternally-deleted outlier. Given this variability, the caveat of litter effects also applicable here, and the single Cre model for the E9.5 embryo data, we maintain the focus of the manuscript on the imprinted genes. The imprinted genes apparently provide the most sensitive system in which to investigate the mechanism of gene regulation by Smchd1, and maternal Smchd1 in particular.

To test association with Polycomb marks, we obtained H3K27me3, H3K36me3 and H3K4me3 peaks from Hanna et al., 2019, in E6.5 epiblast and E7.5 extraembryonic tissue. The *Pcdhb17-22* clustered genes are enriched for H3K27me3, and we detected broad Smchd1-binding on either side in addition to sharper peaks in our newly provided Smchd1 ChIP-seq data from placenta and the existing Smchd1 ChIP-seq in NSCs. Gm8493 was marked with H3K4me3 but not H3K27me3, while Cathepsins, Ceacams and Prolactins did not demonstrate H3K27me3 enrichment in E6.5 epiblast tissue.

We include the histone marks from E7.5 extraembryonic tissue in the depictions of *Smchd1* binding at the *Snrpn*, *Kcnq1* and *Igf2r* clusters in Figure 3—figure supplement 1.

3) Regarding the DNA methylation analyses of maternal SMCHD1 deficiency (RRBS and single embryos): although the authors could not see DNA methylation defects at imprinted loci, what about the rest of the genome? Was there any differentially methylated regions elsewhere?

We appreciate the suggestion to report these analyses. We found little significant differential methylation elsewhere in the genome, which we have now stated in the text.

We have added the following paragraph to the RRBS section:

“Deletion of maternal Smchd1 did not have a large effect on global CpG island methylation. […] The only association with differential expression was the hypermethylation of the CpG island 1.8 kb upstream of Gm45104, downregulated three-fold in placental sample with MMTV-Cre Smchd1 deletion.”

In the single preimplantation embryos, CpG-island and promoter-centric analyses also revealed very few changes in DNA methylation. A genome-wide analysis on 10-kb bins uncovered a bias for hypermethylation but the absolute differences in methylation remained small, and the overall proportion of affected bins small. We have amended the text and present these results in scatter plots (Figure 6—figure supplement 1D-F) and as supplementary tables (Supplementary file 4):

“Furthermore we observe little genome-wide differential methylation in the preimplantation samples: only 8 out 13,840 CpG islands, 45 out of 52,444 promoters, and 872 out of 272,566 10kb bins were differentially methylated, suggesting that maternal Smchd1 contributes very little to overall methylation patterns at the preimplantation stage (Figure 6—figure supplement 1D-F and Supplementary file 4), similar to later in gestation.”

4) Because the authors do not mention it, I suspect that imprinted genes were not among the upregulated genes of the RNA-seq. Could the authors clearly state this in the text? Although the authors report the relative expression of parental alleles, they never provide quantitative values of global expression of these genes showing allelic disturbance. Please provide quantitative RT-PCR of the expression levels of imprinted genes that show the highest degree of reactivation of the normally silent allele in placentae.

We now more fully describe our differential expression analyses genome-wide in the text, as detailed in the answer to point 2. We observe little differential expression with our chosen FDR (5%) and log2-fold-change (1) thresholds. This is not unexpected given that we only observe partial loss of imprinting when using the more powerful allele-specific analyses.

In our study we have used 6 separate RNA-seq datasets to analyse imprinted gene expression (E14.5 placenta oocyte only deletion for 2 Cre lines, E14.5 placenta for heterozygous controls of zygotic deletion, E9.5 embryo oocyte only deletion for 1 Cre line, E14.5 placenta with oocyte and zygotic deletion of Smchd1, E9.5 embryos with oocyte and zygotic deletion of Smchd1). Using these data we can appropriately normalise our data and therefore obtain quantitative measures of gene expression to reliably detect small changes in expression levels. By contrast qRT-PCR is heavily reliant on a small number of housekeeping genes, and is not useful for detecting small expression changes, such as those observed here. Therefore, to clearly show the changes in absolute expression (total and split by allele), we plot these absolute levels in Figure 2—figure supplement 1C for the *Snrpn* cluster (*Igf2r* and *Kcnq1* clusters can also be seen). Along with the MA-plots in Figure 2—figure supplement 1B, these plots demonstrate that the allelic disturbance is explained by a loss of silencing of the repressed allele rather than downregulation of the expressed allele.

5) The authors fail to mention in the text and the figure legends from which ChIP datasets the SMCHD1 binding sites (reported in Figures 2, and 3) are extracted. However, from the Materials and methods section and the legend of Supplementary Figure 3B, it appears that the ChIP-seq was performed in neural stem cells. What is the relevance of reporting the binding sites that SMCHD1 has in neurons relative to expression patterns that were measured in placentae? ChIP-seq analysis in E14.5 placenta (from Smchd1-GFP mouse) is needed.

We agree that ChIP-seq for Smchd1 binding in the placenta is ideal for comparison to our placental data sets. We have now performed ChIP-seq for Smchd1-GFP in a placental sample, and report these data in Figure 3—figure supplement 1. We found that the global enrichment patterns of Smchd1 were very well conserved between the placenta and the NSC ChIP-seq, with the same broad regions detected as enriched by the Enriched Domain Detector software. At a finer scale, the overlap between the peaks called in placenta and NSCs was not as extensive. The resolution of the Smchd1-GFP ChIP-seq in placenta however appeared lower than that of the NSC dataset, therefore we removed the *Smchd1* binding sites from the schematics in Figures 2 and 3, and only show them as a track in Figure 3—figure supplement 1. The text now reads:

“Supporting a role for Smchd1 in regulating the *Kcnq1* cluster, we found Smchd1 enrichment across this cluster as well as the *Snrpn* and *Igf2r-Airn* imprinted clusters in placental samples and somatic cells (Figure 3—figure supplement 1A-C).”

6) The emphasis on the "long-lived" epigenetic effects of SMCHD1 presence in the embryo is overrated. First, effects are mostly seen in the placenta, a short-lived organ whose existence is limited to gestation. Then, we would like to see some late placenta to see whether the effects are maintained, and earlier placenta, to see whether the effects may be reducing with time. Targeted approaches could be used here. If you do not provide these data, please refrain from using "long-lived".

We have refrained from using the term long-lived.

7) The lack of apparent phenotypic effects in the SMCHD1 maternally deficient progenies (at least until weaning age) suggests that the biological relevance of such regulation is weak. This should be clearly acknowledged in the Discussion. Moreover, if SMCHD1 effects are linked to H3K27me3, is there some phenotypic overlap with mouse models of maternal H3K27me3 deficiency (Prokopuk et al., 2018)? In other words, although litter size is normal at weaning, what about the weight? any noticeable difference at birth or any time before weaning?

We have generated additional data on weight of offspring at postnatal day 2, from oocyte deleted mothers, control mothers or from a reciprocal cross. We found that progeny from Smchd1 maternal null females were not significantly different in weight to either set of control litters (wildtype matings from the same strain, or a reciprocal cross). These data are now presented in Figure 1—figure supplement 1B alongside the litter size data (Figure 1—figure supplement 1A). While the genes we do alter are controlled by H3K27me3, they still represent a very limited set of genes compared with all polycomb targets. Moreover, the loss of imprinting we observe usually translates into small absolute log-fold changes, therefore we are not surprised this doesn’t have the same effect as observed by Prokopuk et al. for Eed disruption.

Our data could be viewed as maternal Smchd1 having little biological relevance; however we propose that to make this claim we would need to make more comprehensive analyses of phenotype than weight or litter size in future studies, particularly as the impacts of non-canonical imprints on phenotype are not well known. Here we focused on maternal Smchd1’s role in controlling imprinting, rather than the phenotypic consequences of removing Smchd1. We have noted this limitation of our study in the revised manuscript, and that in the future we will also study the consequences for phenotype. We don’t believe this weakens our study, but instead has kept it focused on the molecular aspects of imprinting control.

8) Please state that other non-canonical imprinted genes were not affected by Smchd1 maternal effect. If the authors do not state it clearly, readers have to perform searches of the supplementary data.

We agree this will be helpful for readers. We have now noted in the Results section “Maternal Smchd1 regulates non-canonical imprinted gene expression” that “No loss of imprinting was observed for the other two non-canonical imprinted genes that retained imprinted expression at this stage of development (*Gab1* and *Slc38a4*).”

9) Genes in the chr7 imprinting cluster (Tssc4 and Ascl2) show clear reactivation (Figure 3). Seeing that imprinting of the neighbouring Cdkn1c and Phlda2 also depends on allelic repressive histone modifications, the authors should report if these are susceptible to LOI and if there are any changes to the Cdkn1c somatic DMR.

Thank you for this suggestion. We have added *Phlda2* to Figure 3A, and added comments on these genes in the Results paragraph dedicated to the *Kcnq1* cluster:

“We observed partial loss of imprinted expression of 2 additional genes in the *Kcnq1* cluster, *Tssc4* and *Ascl2.* […] Again consistent with the *Snrpn* and *Igf2r-Airn* clusters, we observed no DMR hypomethylation at either the germline (*Kcnq1ot1*) or secondary DMRs (*Kcnq1l1* and *2*) in the *Kcnq1* cluster (Figure 3B)”.

10) Please state why E14.5 placentae were analyzed yet E9.5 embryos? Why not at the same developmental age?

Even though this paper focuses on the males, we wanted to dissect embryos at a time when Smchd1 zygotic null female embryos would still be alive and have healthy tissue, i.e. prior to E10. We chose to sample embryos at E9.5 rather than earlier, as it would provide an ample source of tissue. For placentae, our prior work and that of others showed that zygotic Smchd1 is required for imprinting of *Slc22a3*, which is not always imprinted (Zwart et al., Genes Dev 2001). We found more data available on imprinting in mid-gestation placentae than E9.5, and finally, we had the expertise to dissect the embryonic portion of the placenta at E14.5. We have added an explanation to the text:

“We chose to sample embryos at E9.5 as this is before Smchd1 zygotic-null females die; meanwhile mid-gestation placenta is the time and place where many genes display imprinted expression, including those known to be sensitive to loss of zygotic Smchd1. Females were not examined here due to the confounding role of Smchd1 in X chromosome inactivation, which is the focus of a separate project.”

11) On the same matter, please state what cell types were collected and analyzed when referring to "embryonic proportion of the placenta" (Labyrinth, Spongiotrophoblast, giant trophoblast etc?).

Our dissections did not discriminate cell types, but rather were informed by the expression of a GFP transgene transmitted from the father and therefore only present in the embryonic portion of the placenta (see Mould et al., 2013). Such a cross was used to learn the procedure. The tissue dissected was as consistent as possible between samples. We have added comments in the Materials and methods to this effect and noted that based on the known structure of placenta the tissue included labyrinth, spongiotrophoblast and trophoblast giant cells.